



# Surface emergence of glacial plumes determined by fjord stratification

Eva De Andrés[1], Donald A. Slater[2], Fiamma Straneo[2], Jaime Otero[1], Sarah Das[3] and Francisco Navarro[1]

[1] Department of Applied Mathematics, ETSI de Telecomunicación, Universidad Politécnica de Madrid, Madrid, Spain
[2] Scripps Institution of Oceanography, University of California San Diego, La Jolla, CA, USA
[3] Department of Geology and Geophysics, Woods Hole Oceanographic Institution, Woods Hole, MA, USA

*Correspondence to*: Eva De Andrés (eva.deandres@upm.es)

**Abstract.** Meltwater and sediment-laden plumes at tidewater glaciers, resulting from the localized subglacial discharge of surface melt, influence submarine melting of the glacier and the delivery of nutrients to the fjord's surface waters. It is usually assumed that increased subglacial discharge will promote the surfacing of these plumes. Here, at a west Greenland tidewater glacier, we investigate the counterintuitive observation of a non-surfacing plume in July 2012 (a year of record surface melting) compared to the surfacing of the plume in July 2013 (an average melt year). We combine oceanographic observations, subglacial discharge estimates and an idealized plume model to explain the observed plumes' behavior and evaluate the relative impact of fjord stratification and subglacial discharge on plume dynamics. We find that increased fjord stratification prevented the plume from surfacing in 2012, show that the fjord was more stratified in 2012 due to increased freshwater content, and speculate that this arose from an accumulation of ice sheet surface meltwater in the fjord in this record melt year. By developing theoretical scalings, we show in general that fjord stratification exerts a dominant control on plume vertical extent (and thus surface expression), so that studies using plume surface expression as a means of diagnosing variability in glacial processes should account for possible changes in stratification. We introduce the idea that despite projections of increased surface melting over Greenland, the appearance of plumes at the fjord surface could in the future become less common if the increased freshwater acts to stratify fjords around the Ice Sheet. We discuss the implications of our findings for nutrient fluxes, trapping of atmospheric $CO_2$ and the properties of water exported from Greenland's fjords.

## 1 Introduction

Over the last two decades, the rate of mass loss from the Greenland Ice Sheet (GrIS) has quadrupled (Rignot et al., 2011;
Shepherd et al., 2012). Approximately 60% of this ice loss is attributed to increased ice sheet surface melting, while the remaining 40% is due to marine-terminating glacier acceleration and retreat (Jiskoot et al., 2012; Moon et al., 2012) that is thought to result from increased iceberg calving and submarine melting at the glacial fronts (Bamber et al., 2012; van den



Broeke et al., 2009; Enderlin et al., 2014). Thus, understanding processes at the glaciers' fronts is key if we are to understand ongoing changes and to generate future projections.

Among the important processes occurring at the tidewater glacier-ocean boundary, we focus here on buoyant plumes. Buoyant plumes typically occur in localized areas along the glacier front, at times visible on the fjord surface as patches of turbid water (e.g. How et al., 2019; Mankoff et al., 2016). Since they are driven primarily by subglacial discharge deriving from ice sheet surface melting, their appearance is limited mainly to summer (e.g. Motyka et al., 2013; Schild et al., 2016), and, due to the sediments they carry, they control sedimentation rates and distribution in the vicinity of the glacier front (Mugford and

Dowdeswell, 2011). As they rise up the calving front, plumes entrain large volumes of ambient fjord waters, increasing their initial volume by more than an order of magnitude (Mankoff et al., 2016; Mortensen et al., 2013) and acting as the engine of convective-driven circulation in the fjords. Through their vigorous turbulent nature, they enable the transfer of ocean heat to the ice, enhancing submarine melting of the glacial front (Kimura et al., 2014; Sciascia et al., 2013; Slater et al., 2015, 2018; Xu et al., 2013). In addition, they likely affect calving rates by incising undercut notches into the terminus, altering the stress

distribution of ice near the terminus (De Andrés et al., 2018; How et al., 2019; Luckman et al., 2015; O'Leary and Christoffersen, 2013; Schild et al., 2018; Vallot et al., 2018).

Besides the cited physical implications, buoyant plumes also play a key role in important fjord biogeochemical processes. They enrich the uppermost layers of the fjord by upwelling nutrients (e.g. Fe, $NO_3$, $PO_4$, Si) that come primarily from the nutrient rich deep ocean waters, but also from the subglacial bedrock weathering and the ice meltwater (Bhatia et al., 2013;

Cape et al., 2019; Hopwood et al., 2018; Meire et al., 2017). If the nutrient-laden plume reaches the photic zone, the increase in nutrient availability can enhance phytoplankton productivity during the summer season (Hopwood et al., 2018), favoring $CO_2$ trapping in fjord waters (Meire et al., 2015), sustaining important fisheries in Greenland (Meire et al., 2017), and supporting arctic seabird populations (Arimitsu et al., 2012). Alternatively, the turbidity associated with the sediment-laden plumes can also stress benthic ecosystems (Korsun and Hald, 2000) and inhibit light penetration, limiting photosynthesis and,

therefore, phytoplankton productivity (Arimitsu et al., 2012; Meire et al., 2017).

The effect that a buoyant plume will have on the physics and biogeochemistry of the fjord and glacier is sensitive to the vertical extent of the plume in the water column. The vertical extent can influence the distribution of melting along the glacier and therefore the glacier shape (Slater et al., 2017), and the layers that are nutrient enriched (Hopwood et al., 2018). Theoretical considerations suggest that in stratified environments such as glacial fjords, buoyant plumes have two characteristic heights

(List, 1982; Morton et al., 1956). The first is the neutral buoyancy depth (NBD), reached at the depth where the plume density equals the ambient density. The second is the maximum height depth (MHD), situated above NBD, and reached at the depth where the plume vertical velocity decreases to zero (Baines, 2002; Morton et al., 1956). The relationship between these two characteristic heights and the fjord surface determines whether the plume is not visible at the surface, is visible only adjacent



to the glacier, or is visible throughout the fjord (Slater et al., 2016). Theory furthermore suggests that these two characteristic

heights (and thus the vertical extent of the plume) are primarily determined by two factors: the intensity of the subglacial discharge, acting to increase the vertical extent, and the strength of the fjord stratification, acting to decrease the vertical extent (Morton et al., 1956).

Despite the long history of theoretical and modeling work on subglacial discharge plumes, field observations with which to test our understanding remain limited due to the extreme difficulty of obtaining measurements adjacent to tidewater glaciers.

To address this gap, we here present repeat surveys from 2012 and 2013 of a major plume and associated jet at the edge of a mid-sized glacier in central-west Greenland. We find that the plume did not reach the fjord surface in summer 2012, despite this being a year of record surface melting (Tedesco et al., 2013), while the plume did reach the fjord surface in 2013, a year of average melt (Mankoff et al., 2016). We combine our field observations with a plume model to explain these counterintuitive observations, and, more generally, to investigate how plume vertical extent is controlled by subglacial discharge and fjord

stratification. We finally discuss how the vertical extent of plumes may evolve in the future under climate warming.

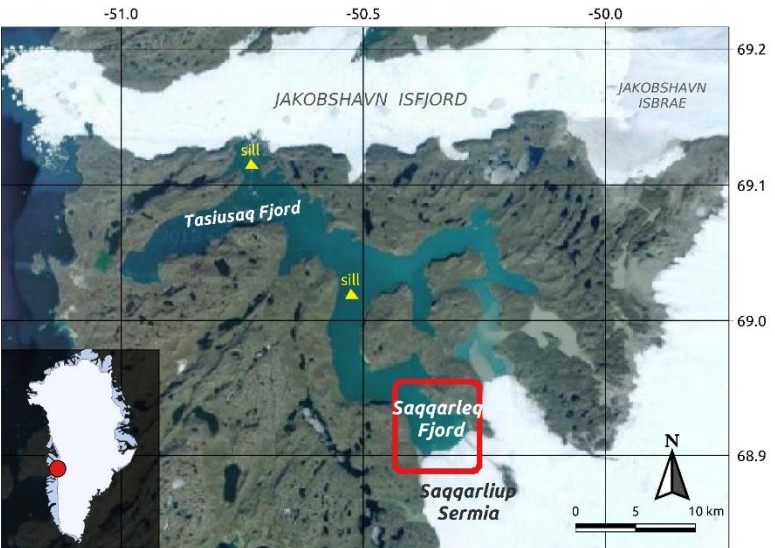

**Figure 1: Location map of Saqqarleq Fjord-Saqqarliup Sermia system (composite image from the U.S. Geological Survey and Google Earth, 2019). The inset shows the location in central-west Greenland.**

## 2 Methods

Saqqarleq Fjord (SF) is the southernmost branch of the intricate system of fjords connected to Jakobshavn Isfjord (JI) over a 125 m-depth sill, in central-west Greenland (Fig. 1). It is a mid-sized fjord and is approximately 35 km long and 6 km wide in





the vicinity of the glacial front (Saqqarliup Sermia, SS), where the depth reaches 150 m (Stevens et al., 2016; Wagner et al.,

2019).SF meets Tasiussaq Fjord (TF) over a sill of 70 m depth, located 15 km from SS terminus (Stevens et al., 2016). SS is a

mid-sized marine-terminating glacier with maximum velocities in summer of 2 m d$^{-1}$ near the calving front (Wagner et al.,

2019), and with an upstream subglacial catchment of $400 \pm 50$ km$^2$ (Stevens et al., 2016).

**2.1 Field data**

Two field surveys were carried out in consecutive summers from 17 to 27 July 2012 and from 24 to 31 July 2013 (Mankoff et

al., 2016; Slater et al., 2018; Stevens et al., 2016; Wagner et al., 2019). In 2012 (2013), a total of 90 (96) CTD (conductivity,

temperature, and depth) profiles were collected using an RBR XR 620 sensor that was calibrated pre- and post-deployment.

CTD stations were distributed along several across-fjord (terminus-parallel) and along-fjord transects (Fig. 2). The CTD

profiles were collected from a small boat and extend from 150 m to 5 km from the glacier terminus. Temperature and salinity

profiles even closer to the glacier front were collected by deploying Sippican xCTDs (expendable CTDs) from a helicopter; 2

such profiles were obtained in 2012 and 12 in 2013. All CTD/xCTD data were pressure-averaged to a resolution of 1 dbar. No

statistical differences were found between CTD/xCTD casts taken on different days (Mankoff et al., 2016) and thus we assume

that properties did not change considerably within either field campaign. Temperature and conductivity values are converted

to conservative temperature ($\Theta$) and absolute salinity ($S_A$) respectively (IOC, SCOR, and IAPSO 2010) using the

thermodynamic equation of state, TEOS-10 (McDougall and Barker, 2011).

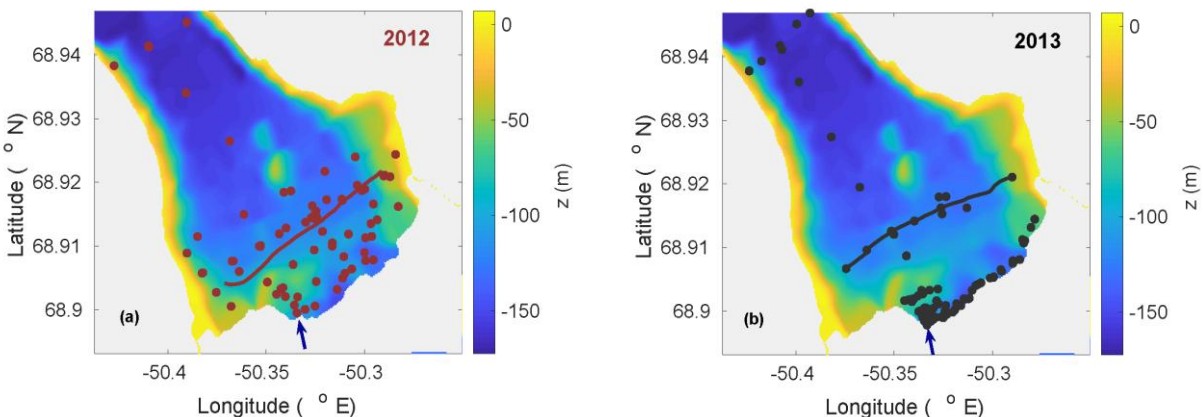

**Figure 2: Bathymetric map of Saqqarleq Fjord area (red rectangle on Fig. 1). CTD cast locations (dots) and ADCP transects (lines) in 2012 (left) and 2013 (right). The location of the main plume is indicated by the blue arrow.**

Parallel to, and at a distance of ~1.5 km from the glacier front, water velocity surveys were performed on July 20 of 2012 (DOY 202) and July 26 of 2013 (DOY 207) (Fig. 2). The observations were obtained from an acoustic Doppler current profiler (ADCP, RDI 300 kHz) mounted on the small boat and binned into 4 m depth bins after removing the ship motion and corrected

for local magnetic declination. ADCP data were processed using CODAS (Common Oceanographic Data Access System) from the University of Hawaii. Data were spatially interpolated by kriging to obtain the cross-sectional (terminus-parallel) contour maps.

Fjord bathymetry was obtained from the shipboard single-beam depth sounder, the shipboard ADCP and the REMUS-100 (remote environmental measuring units) autonomous underwater vehicle (AUV) as described in Stevens et al. (2016) and

Wagner et al. (2019). We also make use of aerial photographs taken from the helicopter in May-June and July of 2012 and 2013 to provide a snapshot of the surface expression of the sediment-laden buoyant plumes.

**2.2 Runoff estimates**

Estimates for subglacial runoff from SS were determined as in Mankoff et al. (2016) and Stevens et al. (2016). Briefly, the SS catchment area was determined based on hydropotential flow routing, governed by SS surface and bed topography (Cuffey

and Paterson, 2010; Stevens et al., 2016). Stevens et al. (2016) determined that SS has three subcatchments each draining through the terminus; in this study, we consider both the SS total catchment ($C_{tot}$) and the largest subcatchment ($C_1$). Once these catchments are defined, subglacial runoff for both 2012 and 2013 is estimated by summing RACMO2.3 surface melting over the catchments (van den Broeke et al., 2009). We make the common assumption that meltwater generated at the glacier surface emerges instantaneously from the glacier grounding line.

**2.3 Buoyant plume model**

Buoyant plume theory is a common tool for developing insight into plume dynamics and the dominant controls on their variability (e.g. Carroll et al., 2015, 2016; Cowton et al., 2016; Jenkins, 2011). The limited information we have on plume geometry suggests a truncated line plume is the most appropriate for plumes driven by subglacial discharge at tidewater glaciers (Fried et al., 2015; Jackson et al., 2017). Therefore, in this study, we use the line plume model of Slater et al. (2016), which is

based on the formulation of Jenkins (2011). The model is used to reproduce the observed plume features and to elucidate the mechanism that suppressed the buoyant plume extent during the record 2012 melt season. We generalize the relative importance of environmental forcings by obtaining a scaling for plume vertical extent in terms of subglacial discharge flux and stratification.





### 2.3.1 Model description

In the plume model, the evolution of the buoyant plume properties (width, vertical velocity, temperature and salinity) along
the vertical tidewater face is described by four ordinary differential equations that conserve the fluxes of mass, momentum,
heat and salt (Slater et al., 2016). The model is steady in time and integrated over the plume cross-section, leaving the along-
flow direction (i.e., $z$) as the only independent variable. The model is closed using constant drag ($2.5 \times 10^{-3}$) and entrainment
(0.09) coefficients, the 75-term non-linear equation of state (TEOS-10, McDougall and Barker, 2011), and three equations

representing the thermodynamic equilibrium at the ice-ocean interface (Holland and Jenkins, 1999), which allows estimation
of the submarine melt rate of the calving front. A full description of the model can be found in Slater et al. (2016).

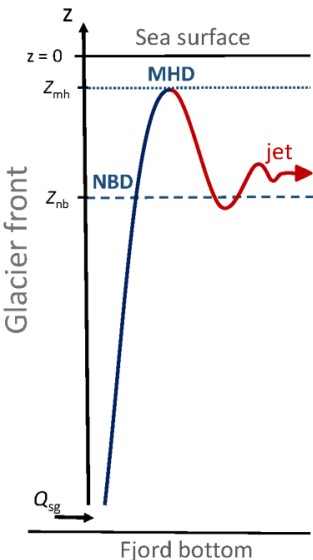

**Figure 3: Schematic of plume characteristic heights - neutral buoyancy depth (NBD) and maximum height depth (MHD) - and the associated jet pathway. Note that the plume model does not represent plume dynamics after the maximum height is reached (red line), but it is expected that the jet will sink to a depth similar to the NBD.**

### 2.3.2 Model experiments

While immersed in stratified environments, vertical plume development is finite and the plume has two characteristic plume

heights (Fig. 3; List, 1982; Morton et al., 1956). The first, NBD, is reached when the plume density equals the ambient density.
From this point, the plume continues upwards due to vertical momentum but slows due to the reversed buoyancy experienced
above the NBD. The plume reaches MHD where the vertical velocity reaches zero (Baines, 2002; Morton et al., 1956; Slater
et al., 2016). Buoyant plume theory does not capture the dynamics of waters in the plume beyond this point, however the



waters are negatively buoyant and will therefore sink as they flow away from the glacier, eventually equilibrating somewhere

near the NBD (Fig. 3; e.g. Carroll et al., 2015). Thereafter, waters in the plume flow horizontally and can be treated as a jet
(Bleninger and Jirka, 2004; Caufield and Woods, 1998; Jirka, 2004).

To analyse the sensitivity of plume vertical extent to subglacial discharge and fjord stratification, we ran the plume model for
each year using ambient fjord conditions constructed from averaging all CTD casts from the given year. We considered a wide
range of subglacial discharge fluxes ($Q_{sg}$, from 10 to 400 m³ s⁻¹, every 10 m³ s⁻¹) and subglacial channel widths ($W$, from 10

to 200 m, every 10 m). We evaluate the model on three principal aspects. First, the fact that according to our field observations,
the plume should surface in 2013 but not in 2012. Second, we compare the modeled plume NBD with the observed depth of
the jet in the water velocity measurements. Third, we compare modeled and observed plume temperature and salinity properties
at the fjord surface.

### 2.3.3 Scalings

After evaluating the model at SF with realistic 2012 and 2013 conditions, we seek to generalise our results by investigating
the scaling of plume vertical extent with subglacial discharge flux and stratification. Stratification may be quantified through
the squared Brunt–Väisälä buoyancy frequency, $N^2$, defined as

$$N^2 = -\frac{g}{\rho_{ref}}\frac{d\rho}{dz} , \qquad (1)$$

where $\rho$ is water density determined from Θ and $S_A$ at depth, $\rho_{ref}$ is the reference density, which, for our purposes, will be that

at the fjord bottom, and $g$ is the gravitational acceleration (with no geographical dependency). To find a scaling, we fit a suite
of plume model results, using non-linear least squares, to a simple curve that takes the form

$$Z = A(N^2)^a Q_{sg}^b , \qquad (2)$$

where $Z$ accounts for the characteristic plume height (either NBD or MHD) in meters, $A$ is a constant of proportionality and $a$
and $b$ are the powers of the scaling. According to the bathymetry and CTD data (see results section), the fjord depth is set to

150 m and divided into two layers: the unstratified bottom layer (from the bottom to 100 m depth) and the stratified top layer
(100 m depth to the sea surface). Given the weak impact of temperature on density, in this exercise, we assume a constant
temperature profile Θ $(z) = 1°$ C (which is in fact close to the real conditions at Saqqarleq, except close to the surface), so that
the stratification is determined solely by salinity gradient. $S_A$ of the bottom layer was held constant at 33.6 g kg⁻¹ while the top
layer is linearly stratified in salinity with a sea surface $S_A$ ranging from 33 to 24 g kg⁻¹, which allows us to analyze stratification



strengths ($N^2$) from 2 to $8 \cdot 10^{-4}$ s$^{-2}$. Runoff ($Q_{sg}$) was varied from 60 to 180 m³ s$^{-1}$ every 20 m³ s$^{-1}$. An identical procedure is

used to find a scaling for submarine melt rate.

## 3 Results

### 3.1 Observations

### 3.1.1 Plume observations

Aerial images show that the main plume at SS was observed at the fjord surface on June 1st 2012 (Fig. 4a) but that by the time

the field campaign was taking place in July 2012, the plume was no longer at the surface (Fig. 4b), and it remained subsurface

for the duration of the 2012 field campaign. During the 2013 field campaign on the other hand, the plume was clearly visible

at the surface throughout the duration of the field campaign (Fig. 4c). Despite the differing surface expression, and as described

below, we know from hydrographic and velocity measurements that the plume and the associated jet were indeed present at

the same location in both years (Mankoff et al., 2016; Stevens et al., 2016).

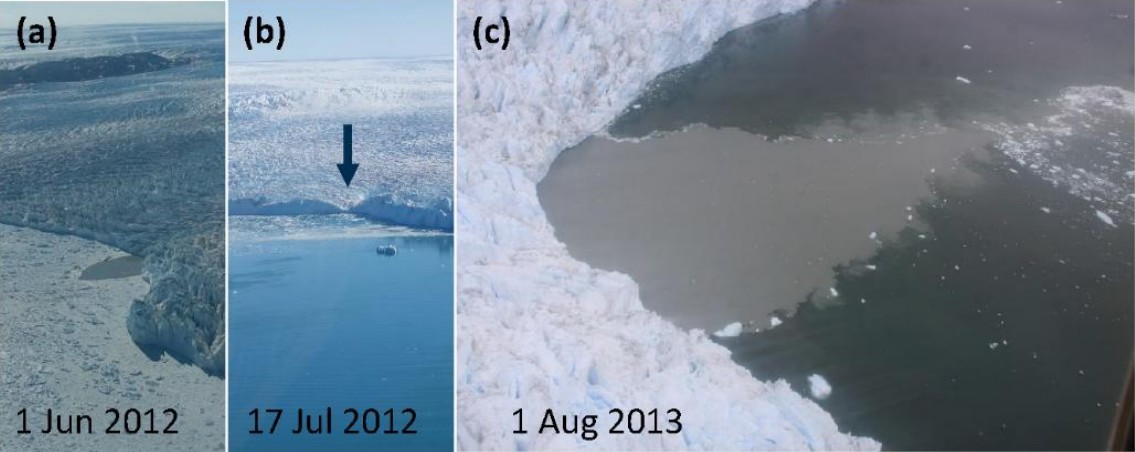

**Figure 4: Aerial images of the main plume at Saqqarliup-Saqqarleq front visible at the fjord surface on a) 1 June 2012 and c) 1 August 2013, but absent on b) 17 July 2012 (the blue arrow indicates plume location; see also Fig. 2).**

### 3.1.2 Fjord structure

Here, we consider the general oceanographic setting that, together with the subglacial discharge, influenced the plume and jet.

In general, the fjord properties were similar in both years with a strongly stratified, warm and fresh, upper 20m layer and a

more weakly stratified deeper layer (Fig. 5). The water column is substantially more stratified in 2012 than 2013, due largely

to fresher conditions in the upper 20m but also a more moderate freshening extending to ~100m depth. Fjord waters in the upper 20 m are also substantially warmer in 2012 than 2013.

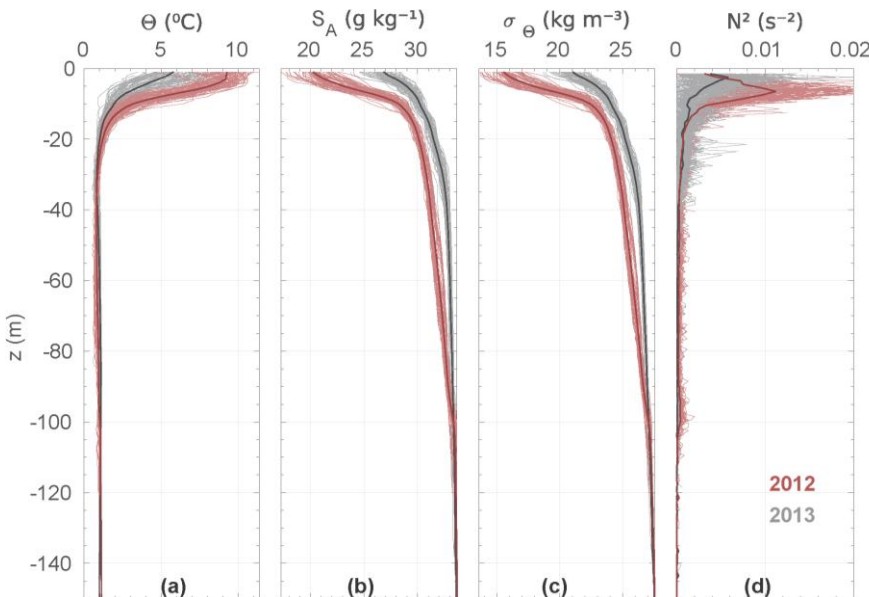

**Figure 5: a) Conservative temperature, b) Absolute salinity, c) sigma-theta (density - 1000 kg m⁻³) and d) squared Brunt-Väisälä frequency profiles (Eq. 1), derived from all CTD casts in Saqqarleq fjord during field surveys in July of 2012 (red) and 2013 (grey). Averaged profiles are shown as darker lines. The water column is characterized in three layers separated by horizontal dashed lines.**

To characterize differences between the years, we first divide the profiles into three layers, according to common characteristics (Fig. 5). The bottom layer, defined from the fjord bottom to $-100$ m, is well mixed in the vertical, has a conservative temperature around 1 °C and absolute salinity of ~ 34.6 g kg⁻¹. Differences observed in this layer between the two years are

negligible. The intermediate layer, from ~20 m to 100 m depth, is also characterized by a temperature of approximately 1 °C and a weak salinity stratification. The salinity gradient within this layer in 2012 is double that of 2013 ($-0.04$ g kg⁻¹ m⁻¹ compared to $-0.02$ g kg⁻¹ m⁻¹). The top layer comprises the uppermost 20 m of the water column and has a strong gradient in both temperature and salinity in both years. The conditions of maximum temperature and minimum salinity occur at the surface. In 2012, surface conditions were warmer (up to 10 °C) and fresher (as low as 17 g kg⁻¹) than in 2013, and the upper

layer was more strongly stratified in 2012 compared to 2013 ($N^2 > 0.11$ s⁻² in 2012 compared to $N^2 < 0.006$ s⁻² in 2013).





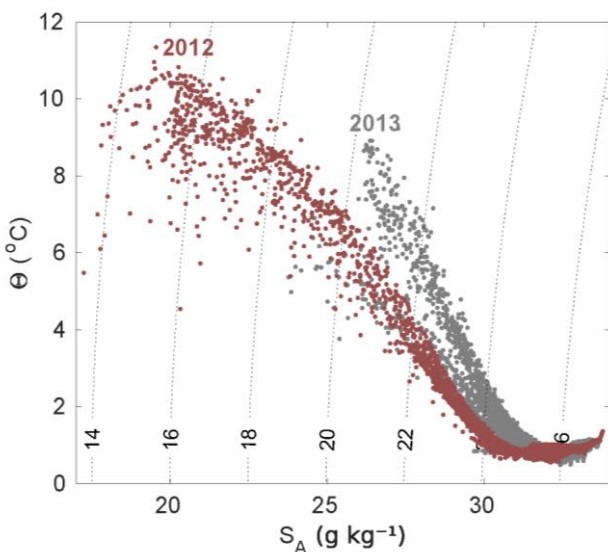

**Figure 6: Conservative temperature vs. Absolute salinity diagram, showing the different water properties in Saqqarleq Fjord during field work in July of 2012 (red) and 2013 (grey). Isopycnals of sigma-density are plotted as near-vertical dotted lines.**

A comparison of $\theta - S_A$ properties of the water masses (Fig. 6) again shows that the decreased salinity in 2012 relative to 2013 was distributed from the intermediate layer ($\sigma_\theta$ of ~ 24-26 kg m$^{-3}$) towards the surface. The near-vertical isopycnals on Fig. 6 result from the dominant effect of salinity on water density within the ranges considered in this study. Thus the freshening of the fjord in 2012 relative to 2013 means that middle and upper layers in 2012 were much lighter than in 2013.

To quantify the additional freshwater in the inner part of the fjord (up to the SF-TF sill, see Fig. 1) in 2012 relative to 2013, we consider the depth range from $z = 0$ m (sea surface) to $z = -100$ m depth (bottom-middle layer interface). We assume the area of the inner part of the fjord to be constant in the vertical, $A_f(z) = A_f \approx 35$ km$^2$, and following Rabe et al. (2011), we calculate the volume of additional freshwater as

$$V_0 = A_f \int_{-100}^{0} \frac{S_{2013} - S_{2012}}{S_{2013}} \, dz, \qquad (3)$$

where $S_{2013,2012}(z)$ are the averaged salinity profiles for the respective years (see Fig. 5). We obtain a freshwater excess of 0.16 km$^3$ (i.e. ~ 0.16 Gt) in 2012 relative to 2013, equivalent to 4.5 m of additional freshwater per unit area of the inner fjord.



### 3.1.3 Plume-driven jets

Velocity data from across-fjord transects approximately ~ 1.5 km from the glacier (Fig. 2) reveal the presence of a jet both in
2012 and 2013 (Fig. 7). The jet is a subsurface-intensified localized region of water flowing away from the glacier, located in
the same spot in the along-front transect, oceanward of the main plume location (Figs. 2, 4 and 7). In 2012, the jet was more
diffuse in the vertical, extending to 35 m depth while in 2013, the jet was confined to the upper 20 m. Maximum velocities of
0.35-0.4 m s$^{-1}$ were found at a depth of 25 m in 2012 and 13 m in 2013. Numerical model of the circulation in this fjord (Slater
et al., 2018) shows that these jets are the horizontal outflow from the main plume (e.g. Fig. 3). Outside of the jets, flow is
generally directed towards the glacier (Fig. 7; Slater et al., 2018).

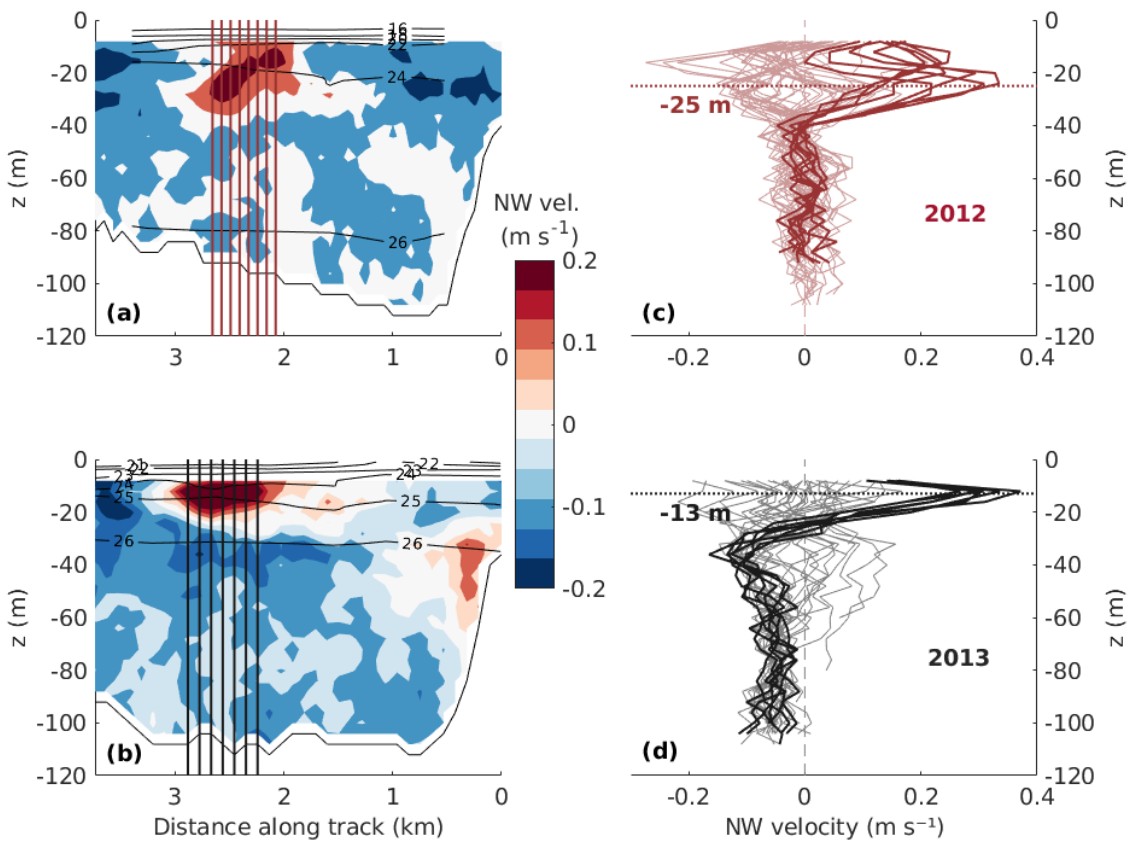


**Figure 7: a) and b) Fjord water velocity transects and c) and d) velocity profiles from ADCP measurements taken in 2012 (top
panels) and 2013 (bottom panels), parallel to and at a distance of 1.5 km from the glacier front (see Fig. 2). Darker profiles in the
right hand panels correspond to the vertical straight lines shown in the left panels, which span the jet.**



## 3.2 Subglacial runoff

One of the main sources of freshwater entering the fjord is the emergence of surface meltwater from the glacier's catchment basin, emerging from beneath the glacier as subglacial runoff (Fig. 8). Glacier surface melting that resulted in substantial runoff began around June 1 (DOY 150) in 2013, and around 10 days earlier in 2012. Runoff is highly variable on daily timescales,

but was generally greater during summer 2012 (average 122 $m^3$ $s^{-1}$) than in summer 2013 (average 92 $m^3$ $s^{-1}$), with a peak runoff in 2012 of ~ 350 $m^3$ $s^{-1}$ far exceeding any value in 2013. During the time period of the fieldwork, mean daily runoff for the total catchment (major subcatchment) was 144 $m^3$ $s^{-1}$ and 132 $m^3$ $s^{-1}$ (113 $m^3$ $s^{-1}$ and 105 $m^3$ $s^{-1}$) in 2012 and 2013 respectively. Considering cumulative summer runoff (Fig. 8), we obtain a total of 0.98 Gt in 2012 and 0.72 Gt in 2013. That is, in 2012 there was additional freshwater runoff input of 0.26 Gt. These differences are consistent with the observation that

2012 was a record melt year in Greenland (Nghiem et al., 2012; Smith et al., 2015; Tedesco et al., 2013).

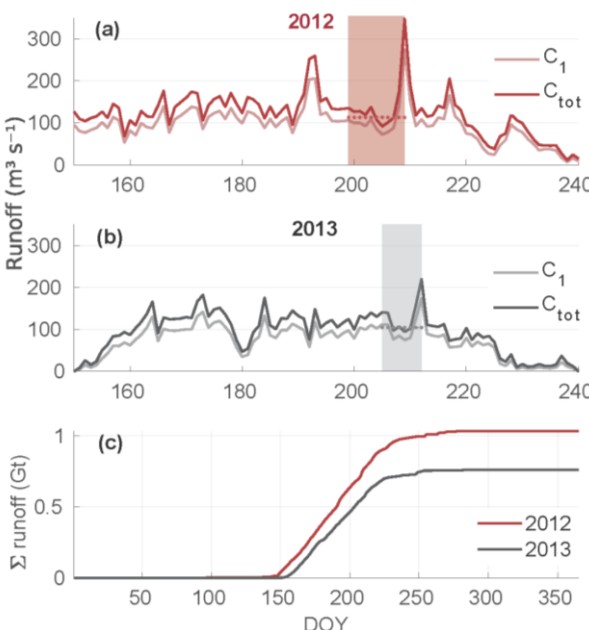

**Figure 8: SS runoff for the total catchment (Ctot, darker lines) and the major subcatchment (C1, lighter lines). Daily runoff estimates are shown from June to August of a) 2012 and b) 2013. Shaded regions comprise the field survey period and the average runoff over this period for C1 is shown inside with a dotted line; c) cumulative runoff volume throughout both years, 2012 (red) and 2013 (dark grey), expressed in Gt.**





### 3.3 Plume modelling

Analysis of the oceanographic data (section 3.1) shows that a plume and the resulting jet were present during both surveys but that their characteristics were different. Specifically, (i) the plume did not reach the fjord surface in July of 2012 while it did in 2013; (ii) fjord conditions were considerably fresher within the intermediate and top layers in 2012 than in 2013; and (iii)

the plume-driven jet was found deeper in 2012 than 2013. Here, we use the line-plume model constrained by the relevant year's bulk oceanographic profiles and forced by different subglacial discharge scenarios to investigate plume behavior. The resulting modeled NBD and MHD for the main plume at SF are shown as a function of the subglacial runoff in Fig. 9. Results are shown for both 2012 and 2013, which differ in their fjord stratification as described above. For a line plume the runoff is prescribed as a runoff per unit width of grounding line ($Q_{sg}/W$), however we also include an axis on Fig. 9 showing the

absolute runoff ($Q_{sg}$) assuming a line plume width of $W = 90$ m, which was suggested by Jackson et al. (2017) to be the most appropriate for the main plume at SS.

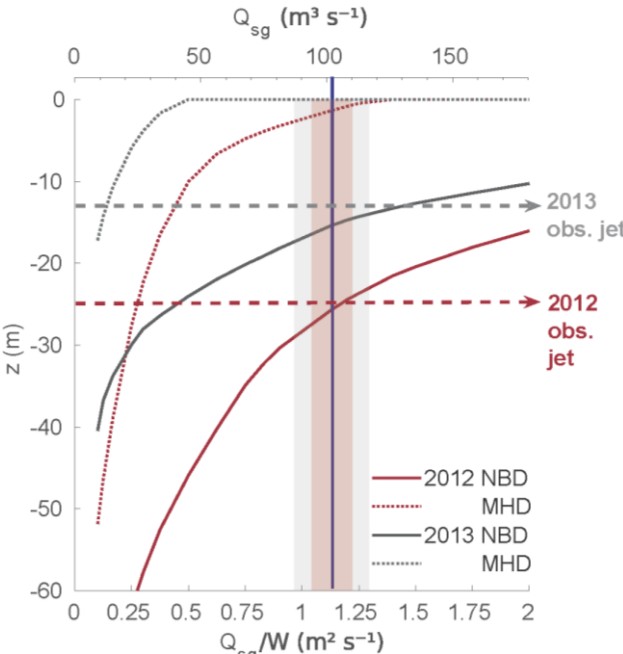

Figure 9: Characteristic plume heights obtained from the line-plume model. NBD (solid lines) and MHD (dotted lines) are obtained for 2012 (red) and 2013 (grey). Dashed horizontal lines mark the depth of the jet core observed from water velocity observations in 2012 and 2013 (Fig. 7). The x-axis at the top represents the subglacial discharge flux applied through a channel width ($W$) of 90 m. The blue vertical line shows the subglacial runoff estimate (from RACMO2.3) averaged over the 5 days prior to the velocity measurements in the fjord, each year (which were approximately the same: 101.7 ± 5.7 m³ s⁻¹ in 2012, and 101.9 ± 13.4 m³ s⁻¹ in 2013). Standard deviations on subglacial discharge are represented by the vertical shaded regions.





We obtained deeper NBD and MHD in 2012 than 2013 for any given $Q_{sg}/W$ ratio (Fig. 9), indicating that the increased
stratification and freshwater content of the fjord in 2012 suppressed the vertical extent of the plume. The NBD remains

subsurface for all of the $Q_{sg}/W$ ratios considered here, indicating that the runoff is insufficient to generate a plume which
would remain at the surface as it flowed down-fjord. The plume reaches the surface (MHD = 0) in 2013 for $Q_{sg}/W$ ratios
higher than ~ 0.4 m2 s-1, while the ratio has to be above ~ 1.3 m2 s-1 for surfacing in 2012 (Fig. 9). Assuming a subglacial
channel width of $W = 90$ m, runoff must exceed ~ 40 m3 s-1 or ~ 120 m3 s-1 for it to surface in 2013 or 2012 respectively.
Our goal is to identify the model parameters that best reproduce our observations of plume surfacing (Fig. 4 in the observations,

MHD in the model) and jet depth (Fig. 7 in the observations, NBD in the model). Following Mankoff et al., 2016, we assume
a subglacial runoff for each year that is averaged over the 5 days prior to the water velocity measurements that identify the jet,
giving $Q_{sg} = 101.7 \pm 5.7$ m³ s⁻¹ in 2012, and $Q_{sg} = 101.9 \pm 13.4$ m³ s⁻¹ in 2013 (Figs. 8 and 9), and we assume a
subglacial channel width of $W = 90$ m in both years (Jackson et al., 2017). With these choices, and as illustrated in Fig. 9 (see
also Fig. 3), we find that (i) the model predicts plume surfacing in 2013 but not 2012 - consistent with observations, and (ii)

the model predicts neutral buoyancy depth which is in reasonable agreement with the observed jet depth. Given that this simple
plume model is able to capture characteristics of the plume and jet in 2012 and 2013, and given that the imposed subglacial
runoff is almost identical between the two years, this confirms that differences in the plumes and jet between the two years are
driven by differences in the stratification of the fjord.

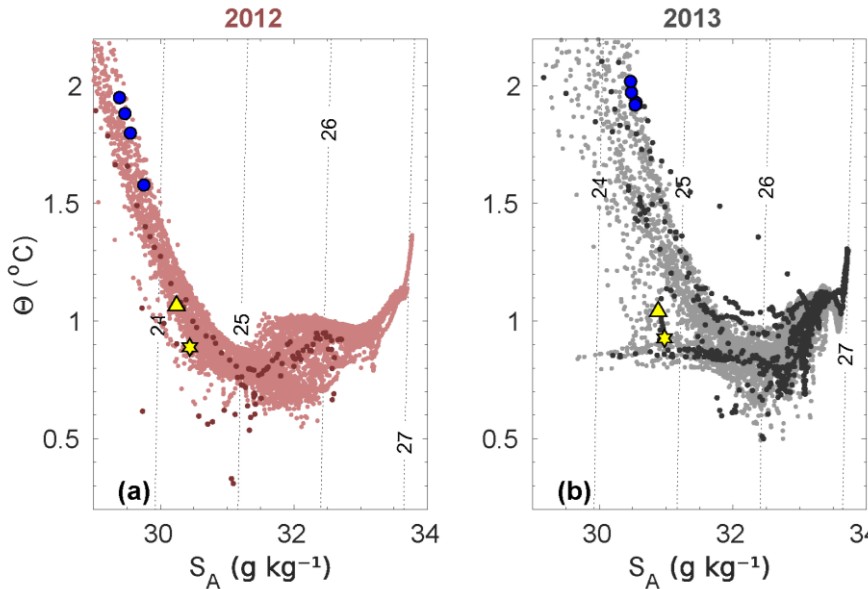

**Figure 10. Conservative temperature and Absolute salinity of Saqqarleq Fjord waters in a) July 2012 and b) July 2013. Light points show CTD measurements while dark dots are xCTD measurements (closest to the plume). Conservative temperature and absolute salinity at the NBD and MHD as predicted by the plume model are shown as a yellow star and triangle, respectively. The blue solid circles represent the water properties in the core of the observed jets in Fig. 7.**



We next consider the modeled plume temperature and salinity at NBD and MHD and compare these with observed properties within the jets flowing down fjord. Plume-model properties at NBD in 2012 are characterized by $S_A$ and $\sigma_\Theta$ of 30.4 g kg⁻¹ and 0.8 ℃, respectively, while they are 31.0 g kg⁻¹ and 0.9 ℃ in 2013 (Fig. 10). The fresher value in 2012 is due to the greater volume of freshwater present in the fjord in 2012 (Figs. 5 and 6), which is entrained into the plume. The properties at MHD (Fig. 10) are warmer and fresher than at NBD, since the plume has by then mixed in some of the warmer and fresher waters

from the upper water column (Figs. 3 and 5). The properties of the jets, ~ 1.5 km from the calving front, are in both years considerably warmer, fresher and lighter than at MHD in the plume (Fig. 10). The outflowing jet is also significantly fresher in 2012 than in 2013.

Lastly, we seek to quantify the relative contribution of runoff and fjord stratification on the vertical extent of the plume in SF through a suite of plume simulations in which we systematically vary runoff and stratification. Given the very good match

with observations (Fig. 9), we use the line plume model and consider a glacier front submerged in water of 150 m depth. To have better control of the stratification parameters we approximate the observed stratification (Fig. 5), by assuming an unstratified bottom layer of 50 m, and a linearly stratified upper layer with fixed thickness of 100 m representing both middle and top layers in SF (see also section 2.3.3). For simplicity we do not separately account for the highly stratified top layer.

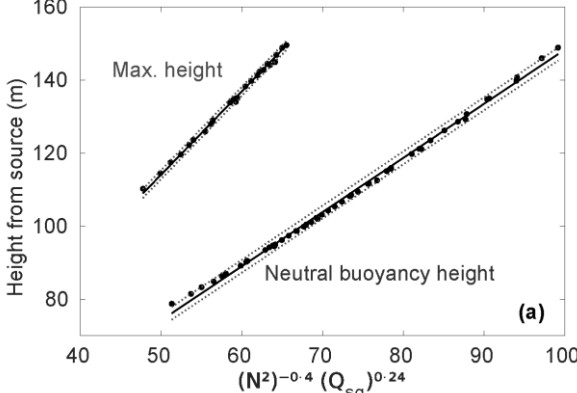
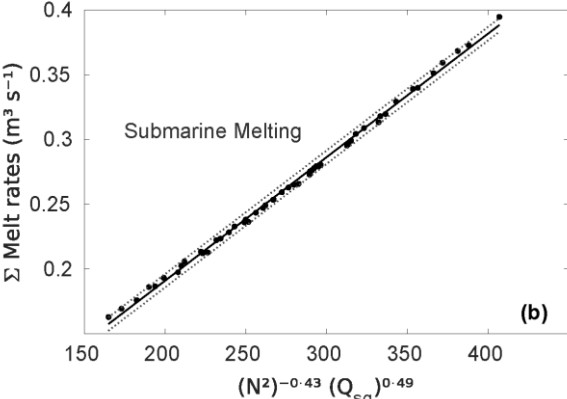

**Figure 11.** Scaling for: a) characteristic plume heights and b) total submarine melt rates from the source to the neutral buoyancy height. Plume model results are plotted by black dots. Straight and dotted black lines represent the fitting curve of Eq. (2) and 95%-confidence bounds, respectively, whose slope and interval bounds values can be found in Table 1.

Figure 11 and Table 1 show the results of fitting curves of the form in Eq. (2) to the results from the plume model. Included are both the plume extents and the vertically integrated submarine melt rate. The power law captures plume vertical extent very well (Fig. 11a), with both neutral buoyancy depth and maximum height scaling with stratification raised to the power



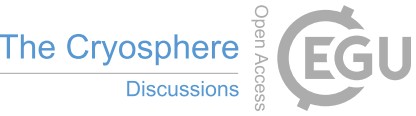

−0.4 and runoff raised to the power 0.24. These scalings are similar to those considered in Slater et al. (2016) - in which the equivalent exponents were −0.5 and 0.3 respectively. Slater et al. (2016) however considered a linear stratification while we

have here considered a two-layer stratification that is more representative of SF. Our results therefore show that power law scalings of the form in Eq. (2) continue to hold in the two-layer case provided small modifications are made to the exponents. It is also notable that the power law scalings for characteristic plume heights (Fig. 11a) perform well even in the absence of the 'point source correction'; an additional term that is often added to the scaling to account for the finite size of the source of subglacial runoff (Slater et al., 2016; Straneo and Cenedese, 2015).

Vertically integrated submarine melt rates (i.e. the total volume of submarine melting resulting from the plume) may also be expressed as a simple function of stratification and runoff (Fig. 11b and Table 1). The stratification exponent is similar to that for the characteristic plume heights. The runoff exponent is however twice that of NBD and MHD, indicating that total melt rate is twice as sensitive to runoff as NBD and MHD. This reflects the fact that submarine melt rate depends on plume velocity, which also scales positively with subglacial runoff.

**Table 1: Results of fitting plume outputs to Eq. 2. The plume outputs presented here are the characteristic plume heights at neutral buoyancy ($Z_{nb}$) and at maximum extent ($Z_{mh}$), and the vertically integrated submarine melt rates from the source to the neutral buoyancy height (Σ SMR).**

| Plume outputs | A | a | b |
|---|---|---|---|
| $Z_{nb}$ | $1.49 \pm 0.09$ | $-0.40 \pm 0.01$ | $0.24 \pm 0.01$ |
| $Z_{mh}$ | $2.28 \pm 0.00$ | $-0.40 \pm 0.01$ | $0.24 \pm 0.01$ |
| Σ SMR | $(1 \pm 0.00) \cdot 10^{-3}$ | $-0.43 \pm 0.01$ | $0.49 \pm 0.01$ |

**4 Discussion**

**4.1 Impact of fjord stratification on plume dynamics in Saqqarleq Fjord**

We have combined a simple plume model with oceanographic data to explain the observation of a discharge plume at SF reaching the fjord surface in 2013 but not in 2012 (Fig. 4), despite 2012 being a record surface melt year at the ice sheet scale. This is consistent with the increased stratification of the fjord in 2012 (Fig. 5) which meant that the characteristic plume heights





(Fig. 3) are significantly deeper in 2012 than in 2013 (Fig. 9). The plume model also suggests that for the plume to reach the surface in 2012, the rate of subglacial discharge would have had to be three times that needed in 2013. The fact that the estimated neutral buoyancy depth is deeper in 2012 (~ 25 m, Fig. 9) than the very fresh layer at the fjord surface (~ 15 m, Fig. 5) suggests that it is not just the fresh surface waters that are influencing plume dynamics but that the differences are also due to the stratification of the intermediate layer.

Given the observed fjord stratification and estimated subglacial discharge, the plume model shows good agreement with our plume and jet observations. The model reproduces the surfacing of the plume in 2013 but not in 2012. The simulated NBD is deeper in 2012 than in 2013, and shows reasonable agreements with the depths at which we observe jets ~ 1.5 km away from the glacier (Fig. 9). Lastly, the temperature and salinity properties of the plume at the fjord surface in 2012 and 2013 lie close to those observed by expendable probes dropped close to the glacier (Fig. 10), indicating that the mixing of the plume and

ambient water is reasonably captured by the model. The model/observations agreement is improved with respect to previous studies of Saqqarleq (Mankoff et al., 2016; Stevens et al., 2016), likely due to their use of a conical rather than line plume model of appropriate width (Jackson et al., 2017).

Our results also show that the observed (or modeled) plume properties - i.e. the properties observed within 150 m of the glacier face which the plume model can reproduce given the observed stratification and estimated discharge - are very different from

those of the waters exported as a jet observed 1.5 km away from the glacier (Fig. 10). The fact that the properties of the jet, in both years, are considerably warmer, fresher and lighter than the observed/modeled plume properties is indicative of significant mixing with the surface waters which must occur as waters from the plume sink and flow away from the glacier. We stress that plume model does not include this dilution - something that must be taken into account both in interpreting observations taken farther than the 'plume distance' from any glacier face (presumably roughly equal to the depth at the grounding line)

and/or in extrapolating plume observations/properties away from the glacier. More complex models are needed to capture this mixing and export (e.g. Slater et al., 2018).

Despite good agreement between model and observations in the plume characteristics, a number of key assumptions are worth commenting on. We have assumed that meltwater from the glacier surface emerges from the grounding line instantaneously, so that estimated daily surface melting can be equated to daily subglacial discharge. Although this is a widespread assumption

in glacier-fjord studies (Mankoff et al., 2016; Slater et al., 2018; Stevens et al., 2016), it is a simplification because a number of hydrological processes will act to delay this meltwater, including storage of water in supra- and sub-glacial lakes, and the finite transit time of meltwater along the ice sheet surface and bed (Fountain and Walder, 1998). This delay is likely to be significantly longer, perhaps even weeks, at the beginning of the melt season when there is still a significant snowpack and the subglacial drainage system may be inefficient (De Andrés et al., 2018; Campbell et al., 2006; Cowton et al., 2013; Schild et

al., 2016). As the melt season progresses, drainage becomes more efficient with subglacial transit velocities exceeding 1 m s⁻



[1] (Cowton et al., 2013) so that by late July when our field seasons took place, surface meltwater likely emerges from the grounding line as subglacial discharge rather rapidly, supporting our assumption. Nevertheless, uncertainty on meltwater transit time results in uncertainty in the magnitude of subglacial discharge, however we do not believe this is sufficient to modify our conclusions.

Another source of uncertainty is the width of the subglacial channel delivering discharge into the fjord. Following Jackson et al. (2017), we have considered a channel of fixed width equal to 90 m. It is however expected that channel width growths with subglacial discharge due to increased melting of the channel's walls (Greenwood et al., 2016; Lliboutry, 1983). It is therefore plausible that due to the overall higher subglacial discharge in 2012 (Fig. 8), the main discharging channel at SF was larger in 2012 than 2013. A larger channel could contribute to the plume not surfacing in 2012; if the discharge is more laterally spread

the resulting plume is less intense and does not attain the same vertical extent. The plume model nevertheless shows that the channel width would have to change by a factor of ~ 3 to assume equal importance to the differing fjord stratification. Since channel theory suggests this is unlikely (e.g. Slater et al., 2015), we have here focused on the impact of fjord stratification. We lastly generalized our results by using the plume model to fit a scaling between stratification ($N^2$), subglacial discharge ($Q_{sg}$), and characteristic plume heights NBD and MHD. We found that both characteristic plume heights scaled

with $N^2$ raised to the power $-0.4$ and $Q_{sg}$ raised to the power 0.26 (Fig. 11a), which are similar to those obtained by Slater et al. (2016). This means that a doubling of subglacial runoff will increase plume vertical extent (NBD and MHD) by 18% while a doubling of stratification decreases plume vertical extent by 25%. While the net impact on plume vertical extent depends on the intrinsic variability of runoff and stratification, this scaling taken together with our observations shows that stratification plays a dominant role in setting plume vertical extent. In contrast, a doubling of runoff increases total submarine melting by

40% while a doubling of stratification decreases total submarine melting by 26% (Fig. 11b). For submarine melting therefore, stratification is not dominant, but still plays an important role that is worth considering in bulk submarine melt rate parameterisations.

**4.2 Controls on fjord stratification**

Large fjords in Greenland exhibit a characteristic vertical structure where cold and fresh polar water (PW) occupies the surface

layer and extends to as deep as 200 m, overlying warmer and saltier Atlantic Water (AW) at depth (Straneo et al., 2012; Straneo and Cenedese, 2015). SF is relatively shallow, having a maximum depth of 230 m, and is separated from the open ocean by sills at 70 m to Tasiusaq Fjord and 125 m to Ilulissat Icefjord (Fig. 1). As such we do not see AW in Saqqarleq Fjord, rather we see cooler Ilulissat Icefjord waters (IIW, Stevens et al., 2016). Our CTD profiles from SF show three well-differentiated layers during summer (Fig. 5). The bottom layer (below 100 m depth) is homogeneous with properties likely controlled by

shear mixing over sills (Carroll et al., 2017; Gladish et al., 2015a). Our data showed no interannual variability between 2012



and 2013 within this layer. In contrast, the intermediate (100 m to 20 m depth) and top (above 20 m depth) layers showed interannual variability with an additional 4.5 m of freshwater present in these layers in 2012 compared to 2013 (Eq. (3) and Fig. 5).

By analogy with other fjords around Greenland, water properties in SF are expected to experience strong seasonal variability
as a consequence of increased glacial freshwater inputs and solar radiation during summer (Jackson et al., 2014; Schild et al., 2016; Sciascia et al., 2013; Straneo et al., 2011). We have focused on contrasting plume dynamics between July 2012 and July 2013, but in fact we also observed the plume at the fjord surface in early June 2012, when runoff is low (Figs. 4 and 8). We do not have any records of fjord properties in early June 2012, but we suspect the plume was able to surface due to a relatively unstratified water column at the beginning of the melt season. The strong stratification and the subsurface trapped plume in
late July 2012 suggests seasonal variability in fjord stratification with the fjord becoming more stratified as the melt season progresses.

The additional freshwater in the fjord in July 2012 relative to July 2013 amounts to 0.16 Gt when summed over the inner part of SF (i.e. the region shown in Fig. 2). This could be accounted for by the high subglacial discharge in 2012 which, by the end of the melt season, exceeded that from 2013 by 0.26 Gt. We do not here attempt a rigorous freshwater budget, which would
account for additional freshwater sources and sinks such as the formation and melting of sea ice, melting of the calving front and icebergs, land runoff, and freshwater import and export from the fjord. Rather we suggest that due to the strong zones of recirculation observed and modeled in SF during summer (Slater et al., 2018), it is plausible that a significant fraction of the additional freshwater in 2012 remained in the inner fjord long enough to freshen the water column, leading to a stronger stratification and inhibiting the vertical extent of the plume in July 2012 compared to June 2012 and July 2013. The implication
is that the glacier itself impacts the stratification of the fjord which, in turn, will have an impact on glacier/ocean exchanges and on where/how the meltwater is exported (Curry et al., 2014; Gladish et al., 2015a, 2015b; Oliver et al., 2018; Straneo et al., 2011).

The increased freshwater content of the fjord in 2012 is not limited to the surface layer, instead extending to 100 m depth (Fig. 5). Precipitation, sea ice melting and land runoff would most strongly affect the near-surface, and would have to be mixed
downwards to significantly impact properties at depth. Therefore the increased freshwater content at depth is more likely to have a glacial origin; either the melting of large, deep-keeled icebergs (Enderlin et al., 2016; Moon et al., 2018), melting of the calving front itself (Slater et al., 2018; Wagner et al., 2019), or the trapping of subglacial discharge plumes below the surface (Fig. 4; Stevens et al., 2016). Considering the last point, secondary discharge channels with weaker plumes that find neutral buoyancy at greater depths (Slater et al., 2018) may play an important role in setting the seasonal fjord stratification.
Equally, temporal variability in subglacial discharge of the main plume, resulting in periods where the plume reaches neutral buoyancy at depth, may drive freshening of the fjord and feedback on the dynamics of the plume. Overall we are suggesting



that high surface melting through the melt season in 2012 may have freshened the fjord, driving increased fjord stratification and leading to the suppression of the plume later in the melt season.

## 4.3 Wider impacts of glacier-fjord coupling

We have provided evidence that surface melting of a marine-terminating glacier, and the associated subglacial discharge, together with the fjord's stratification exert a strong control on the dynamics of subglacial discharge plumes with implications for melting of the glacier face and export of meltwater. We have also speculated that part of the differences between 2012 and 2013 in SF are due to the impact of the extreme surface melt of 2012 on the fjord raising the possibility of feedbacks between surface melt, submarine melt and export. Considering that, under a high greenhouse gas emissions scenario (RCP8.5),

subglacial runoff may increase by as much as a factor of 6 by the end of the century (Slater et al., 2019), it is possible that fjords will become increasingly stratified. Since stratification has proven such an important determinant of plume dynamics in this study, it is possible that despite the increased buoyancy provided by increased subglacial discharge, plumes may reach the fjord surface less over the coming century. This may decrease our ability to observe and monitor plumes based on their surface expression, which has served as a basic but important observation for studies of fjord processes and subglacial hydrology

(Schild et al., 2016; Slater et al., 2017).

From a biogeochemical perspective, a suppression of the vertical extent of plumes driven by increased fjord stratification could limit the upwelling of nutrients in deep water masses and from subglacial bed weathering (Cape et al., 2019; Hopwood et al., 2018; Meire et al., 2017). Many of these nutrients act as a limiting factor for the primary productivity (phytoplankton) within the photic zone (Cape et al., 2019). Therefore, in contrast to some expectations (Bhatia et al., 2013), an increase in ice sheet

surface melting could have a negative impact on the productivity of fjords. Considering that primary producers are the base of the pelagic ecosystem, a decrease in the productivity of fjord waters could negatively impact fisheries and bird populations (Arimitsu et al., 2012; Meire et al., 2017). It has also been observed that the surface layer of the fjord waters (in contact with the atmosphere) is undersaturated in $CO_2$ during the summer. Around 28% of the uptake is attributed to the input of glacial waters and ~72% to primary producers (Meire et al., 2015). Therefore, a reduction of these organisms together with the

subsurfacing of glacial waters could decrease the ability of the fjords to act as an atmospheric $CO_2$ sink.

Regarding the potential implications on the submerged icefront melting, our scalings show that stratification does indeed suppress melting of the calving front within the plume through dampening of its vertical velocity and extent. However, increased subglacial discharge has a stronger influence on melting through increasing the vertical velocity, and therefore submarine melt rates are likely to increase in response to increased ice sheet surface melting though their vertical reach may

be diminished potentially leading to undercutting.



Lastly, stratification likely impacts on circulation more widely in the fjord, though this is beyond what we can quantify with a simple plume model. Our oceanographic observations of the jet show that due to increased stratification in 2012 compared to 2013, the jet that carries plume waters away from the glacier is deeper (Fig. 7) and fresher (Fig. 10) in 2012 than in 2013. These waters are subsequently exported from the fjord to the continental shelf where they may impact shelf properties (Luo et al., 2016), primary productivity (Arrigo et al., 2017; Oliver et al., 2018) and potentially the larger-scale ocean circulation (Böning et al., 2016; Saenko et al., 2017). Our observations suggest that in the future, increased ice sheet surface melting may stratify Greenland's fjords and modify the depth and properties of waters that are exported to the shelf. Further observations and modeling would be needed to better understand how these processes will evolve in the future.

## 5 Conclusions

This study began with the counterintuitive observation of a surfacing subglacial discharge plume in Saqqarleq Fjord in late July 2013 (an average melt year) but a subsurface trapped plume during late July 2012 (a record melt year). Increased subglacial discharge acts to drive a stronger plume that, in the absence of other factors, will have a greater vertical extent and probability of reaching the fjord surface. By combining oceanographic observations together with a plume model we have shown that the difference between the two years can be explained by the increased freshwater content of the fjord in 2012 relative to 2013, resulting in stronger fjord stratification and a suppression of the vertical extent of the plume. As such, seasonal and interannual variability in fjord stratification has a strong impact on the vertical extent of subglacial discharge plumes at tidewater glaciers. We suggest that the increased stratification and freshwater content of the fjord in 2012 compared to 2013 is driven by the glacier itself. In particular, strong ice sheet surface melting throughout the summer of 2012, delivered to the fjord as subglacial discharge, may have gradually accumulated freshwater in the fjord and increased stratification, providing a negative feedback on plume vertical extent.

Observations of the horizontal jet emanating from the plume in 2012 and 2013 show that the jet is deeper and more diffuse in 2012, and that it carries fresher and lighter water. This interannual difference is consistent with results from the plume model, in which the simulated neutral buoyancy depth of the plume proves a good estimator of the depth of the jet, and shows once more that the driver of the observed differences is the increased stratification of the fjord in 2012. Since waters in the jet are those which will be exported from the fjord, variability in fjord stratification will impart variability on the depth and properties of waters exported from the fjord to the open ocean. We also showed, however, that the properties of waters exported from the glacier/ocean boundary in the jet approximately 1.5 km from the ice front cannot be described by a plume model. Instead, the jet is carrying strongly diluted plume waters through mixing with surface waters. This means that plume models or near-ice front properties are not representative of properties of the meltwater/ambient water mixture.



We generalized our results by fitting a scaling for plume vertical development and total submarine melting in terms of fjord stratification ($N^2$) and subglacial discharge ($Q_{sg}$). We found that plume vertical extent is proportional to $(N^2)^{-0.4}Q_{sg}^{0.24}$ while total submarine melting is proportional to $(N^2)^{-0.43}Q_{sg}^{0.49}$. These highlight the important role played by fjord stratification, and the subglacial discharge flux, in the dynamics and impacts of subglacial discharge plumes.

Looking to the future, we are likely to see increased surface melting of the ice sheet in response to climate warming. Our
results suggest that through increasing the stratification of glacial fjords, it is possible that this melting may suppress rather than promote the vertical extent of plumes and their presence at the fjord surface. This may limit our ability to monitor plumes remotely, reduce the delivery of nutrients to the photic zone, and modify the depth and properties of waters exported from the ice sheet to the ocean. Further observations and modeling are needed to better understand how the stratification of fjords and impacts on physical and biological systems may evolve in the future.

**Author Contribution**


EDA, DS and FS designed the research. FS and SD collected field observations. EDA processed field and runoff data. DS provided the original model code and EDA adjusted the code to this work. EDA performed the analysis and all the authors contributed to the discussion. EDA wrote the original text of the paper with input of all other authors.

**Acknowledgements**

This research was funded by the European Union's Horizon 2020 research and innovation programme under grant agreement No 727890 and by the Spanish State Plan for Research and Development under grants CTM2014-56473-R and CTM2017-84441-R (MICIU/AEI/FEDER, UE). Eva De Andrés is supported by the Spanish Ministry of Education with the PhD studentship FPU14/04109. Fiamma Straneo and Donald Slater would like to acknowledge WHOI's Ocean and Climate Change Institute for funding the fieldwork and NSF 1418256 for funding the analysis. We would also like to thank Dan Torres, James
Holte, Jeff Pietro, Clark Richards, Laura Stevens, Rebecca Jackson, Ken Mankoff, Amy Kukulya, Hanumant Singh, Robin Littlefield, Al Plueddemann, and Ove Villadsen, and colleagues from Illimanaq, for their instrumental role in collecting the data, and in followup discussions and Till Wagner for discussions about the paper.



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



**Table 1: Results of fitting plume outputs to Eq. 2. The plume outputs presented here are the characteristic plume heights at neutral buoyancy ($Z_{nb}$) and at maximum extent ($Z_{mh}$), and the vertically integrated submarine melt rates from the source to the neutral buoyancy height ($\Sigma$ SMR).**

| Plume outputs | A | a | b |
|---|---|---|---|
| $Z_{nb}$ | $1.49 \pm 0.09$ | $-0.40 \pm 0.01$ | $0.24 \pm 0.01$ |
| $Z_{mh}$ | $2.28 \pm 0.00$ | $-0.40 \pm 0.01$ | $0.24 \pm 0.01$ |
| $\Sigma$ SMR | $(1 \pm 0.00) \cdot 10^{-3}$ | $-0.43 \pm 0.01$ | $0.49 \pm 0.01$ |

625



**Figure 2: Location map of Saqqarleq Fjord-Saqqarliup Sermia system (composite image from the U.S. Geological Survey and Google Earth, 2019). The inset shows the location in central-west Greenland.**



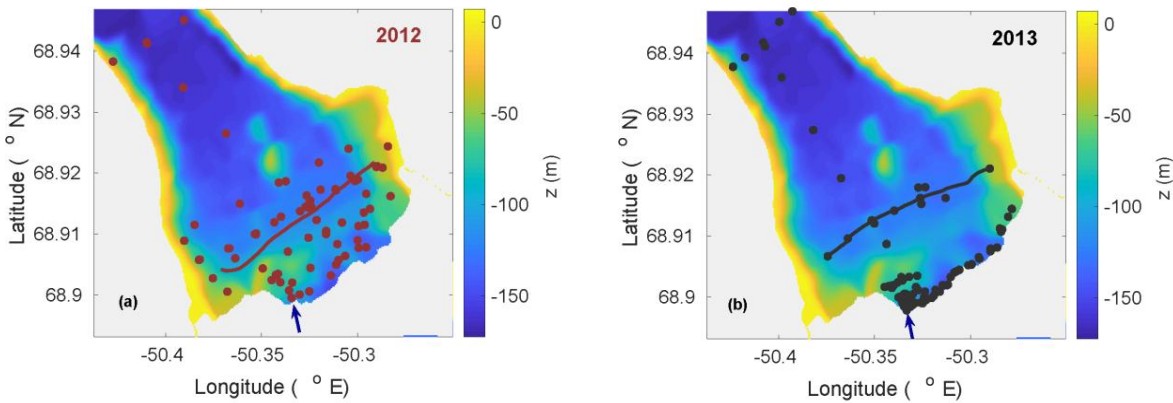

**Figure 2: Bathymetric map of Saqqarleq Fjord area (red rectangle on Fig. 1). CTD cast locations (dots) and ADCP transects (lines) in 2012 (left) and 2013 (right). The location of the main plume is indicated by the blue arrow.**




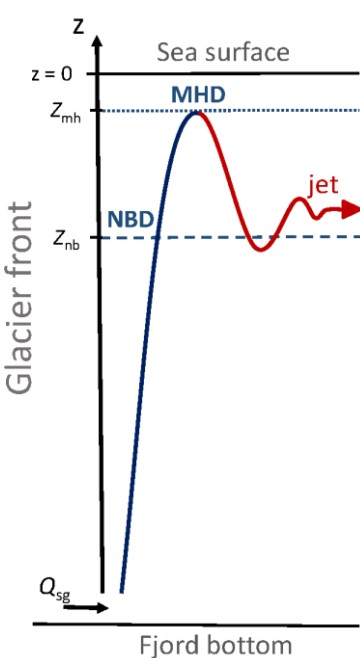

**Figure 3: Schematic of plume characteristic heights - neutral buoyancy depth (NBD) and maximum height depth (MHD) - and the associated jet pathway. Note that the plume model does not represent plume dynamics after the maximum height is reached (red line), but it is expected that the jet will sink to a depth similar to the NBD.**





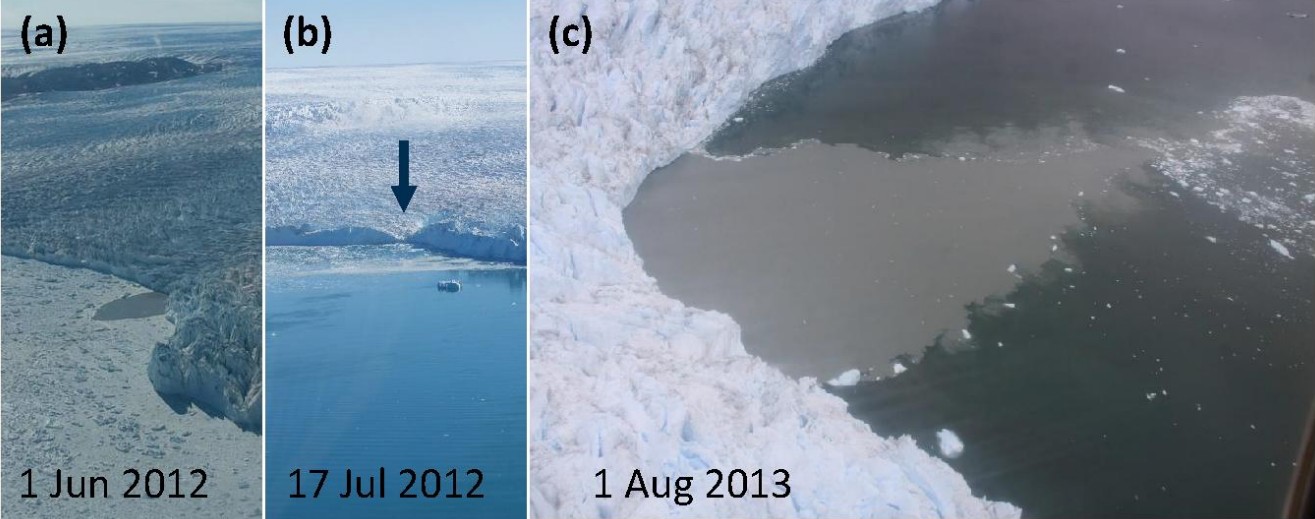

**Figure 4: Aerial images of the main plume at Saqqarliup-Saqqarleq front visible at the fjord surface on a) 1 June 2012 and c) 1 August 2013, but absent on b) 17 July 2012 (the blue arrow indicates plume location; see also Fig. 2).**

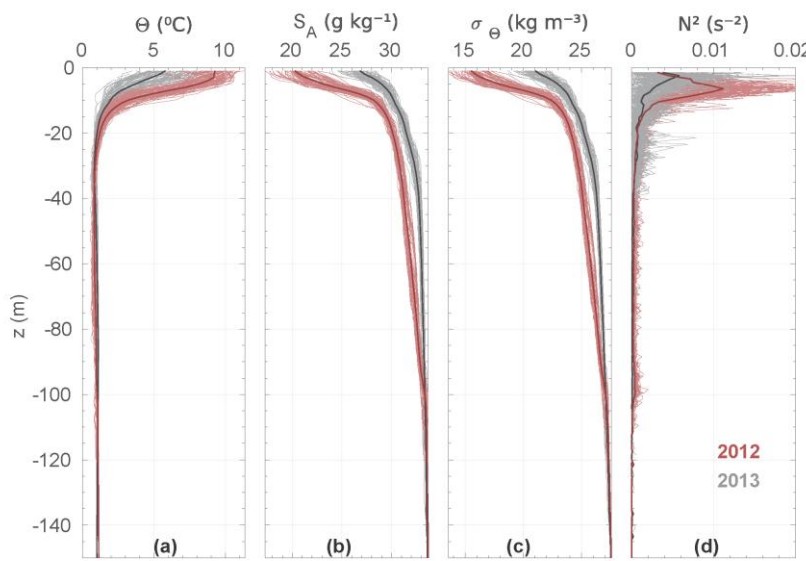

Figure 5: a) Conservative temperature, b) Absolute salinity, c) sigma-theta (density - 1000 kg m⁻³) and d) squared Brunt-Väisälä frequency profiles (Eq. 1), derived from all CTD casts in Saqqarleq fjord during field surveys in July of 2012 (red) and 2013 (grey). Averaged profiles are shown as darker lines. The water column is characterized in three layers separated by horizontal dashed lines.



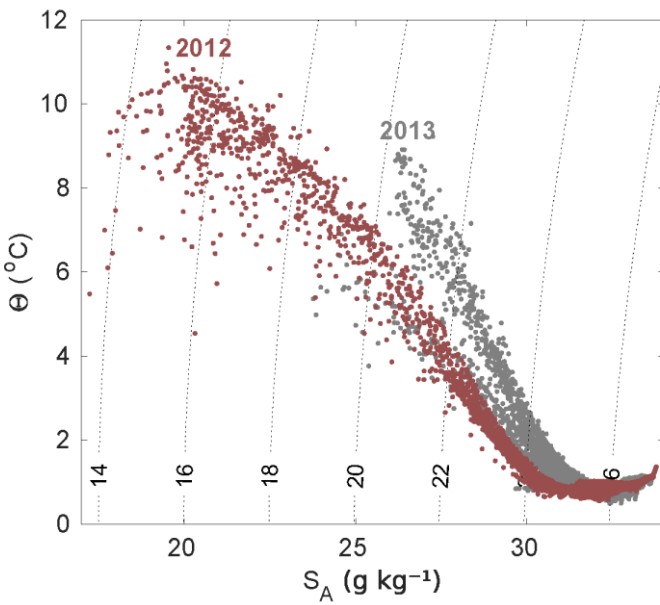

**Figure 6: Conservative temperature vs. Absolute salinity diagram, showing the different water properties in Saqqarleq Fjord during field work in July of 2012 (red) and 2013 (grey). Isopycnals of sigma-density are plotted as near-vertical dotted lines.**



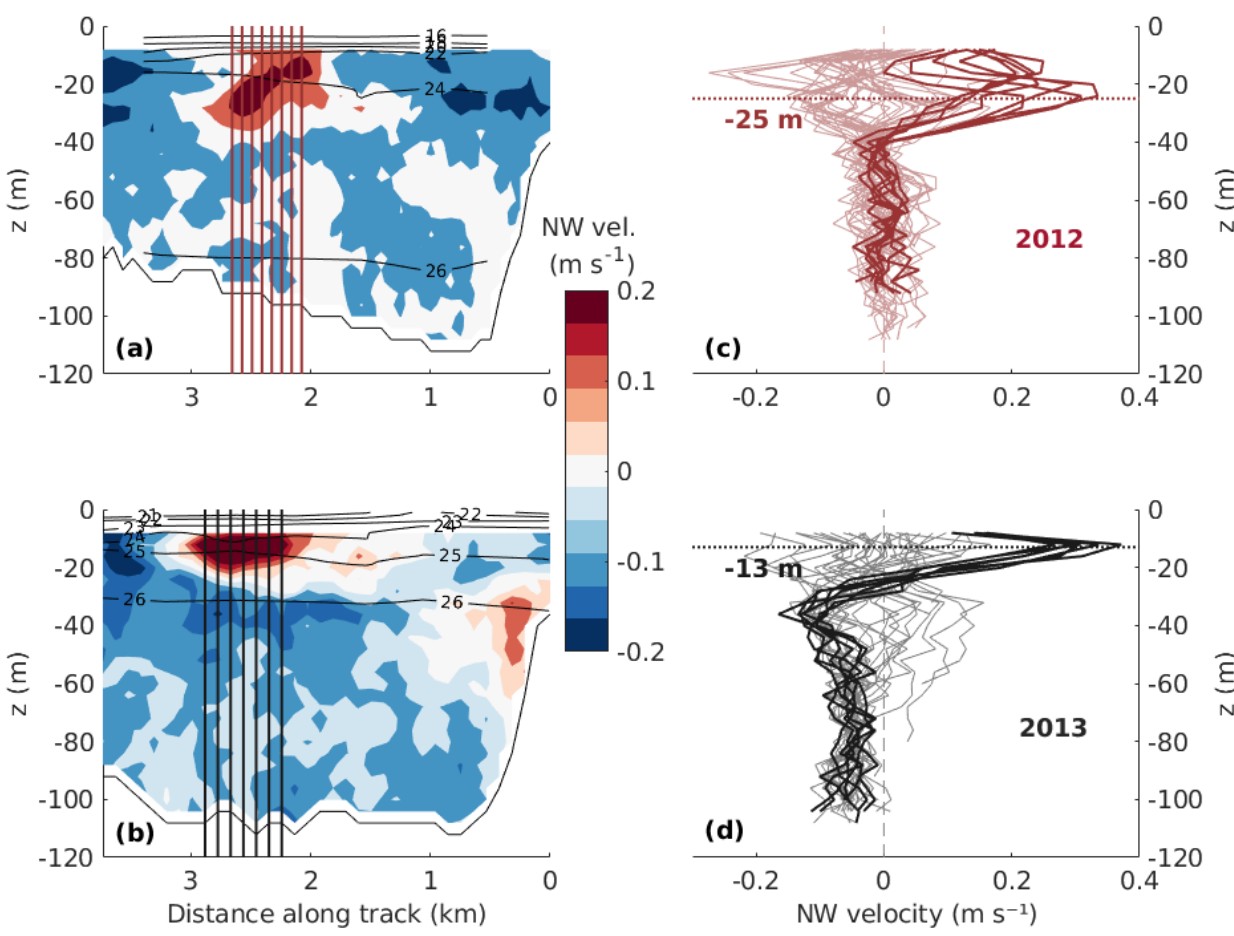

**Figure 7: a) and b) Fjord water velocity transects and c) and d) velocity profiles from ADCP measurements taken in 2012 (top panels) and 2013 (bottom panels), parallel to and at a distance of 1.5 km from the glacier front (see Fig. 2). Darker profiles in the right hand panels correspond to the vertical straight lines shown in the left panels, which span the jet.**



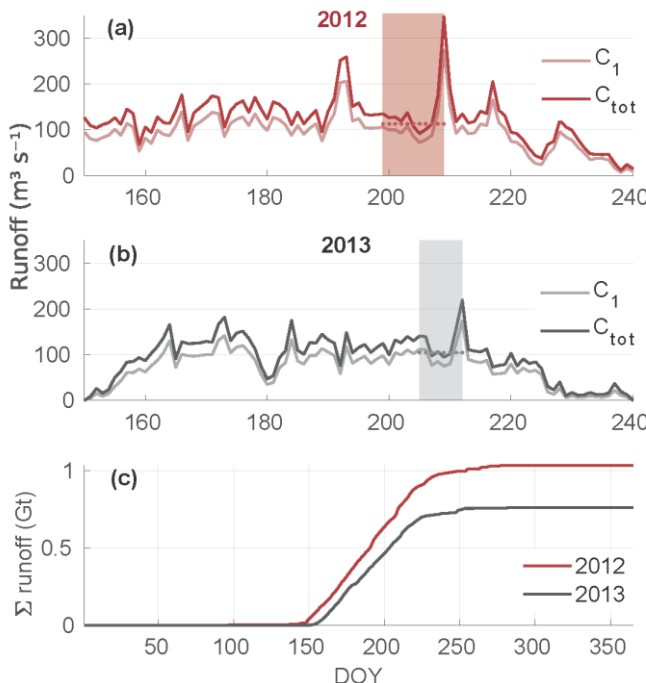

**Figure 8: SS runoff for the total catchment (Ctot, darker lines) and the major subcatchment (C1, lighter lines). Daily runoff estimates are shown from June to August of a) 2012 and  b) 2013. Shaded regions comprise the field survey period and the average runoff over this period for C1 is shown inside with a dotted line; c) cumulative runoff volume throughout both years, 2012 (red) and 2013 (dark grey), expressed in Gt.**





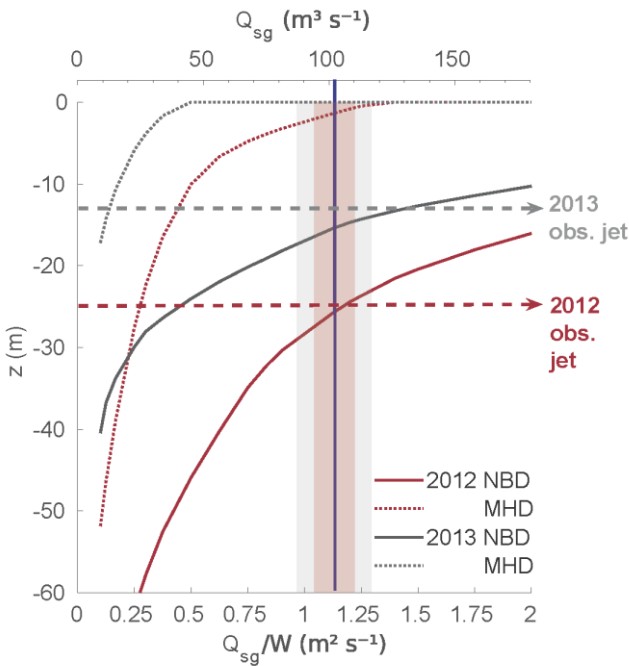

**Figure 9: Characteristic plume heights obtained from the line-plume model. NBD (solid lines) and MHD (dotted lines) are obtained for 2012 (red) and 2013 (grey). Dashed horizontal lines mark the depth of the jet core observed from water velocity observations in 2012 and 2013 (Fig. 7). The x-axis at the top represents the subglacial discharge flux applied through a channel width ($W$) of 90 m. The blue vertical line shows the subglacial runoff estimate (from RACMO2.3) averaged over the 5 days prior to the velocity measurements in the fjord, each year (which were approximately the same: 101.7 ± 5.7 m³ s⁻¹ in 2012, and 101.9 ± 13.4 m³ s⁻¹ in 2013). Standard deviations on subglacial discharge are represented by the vertical shaded regions.**



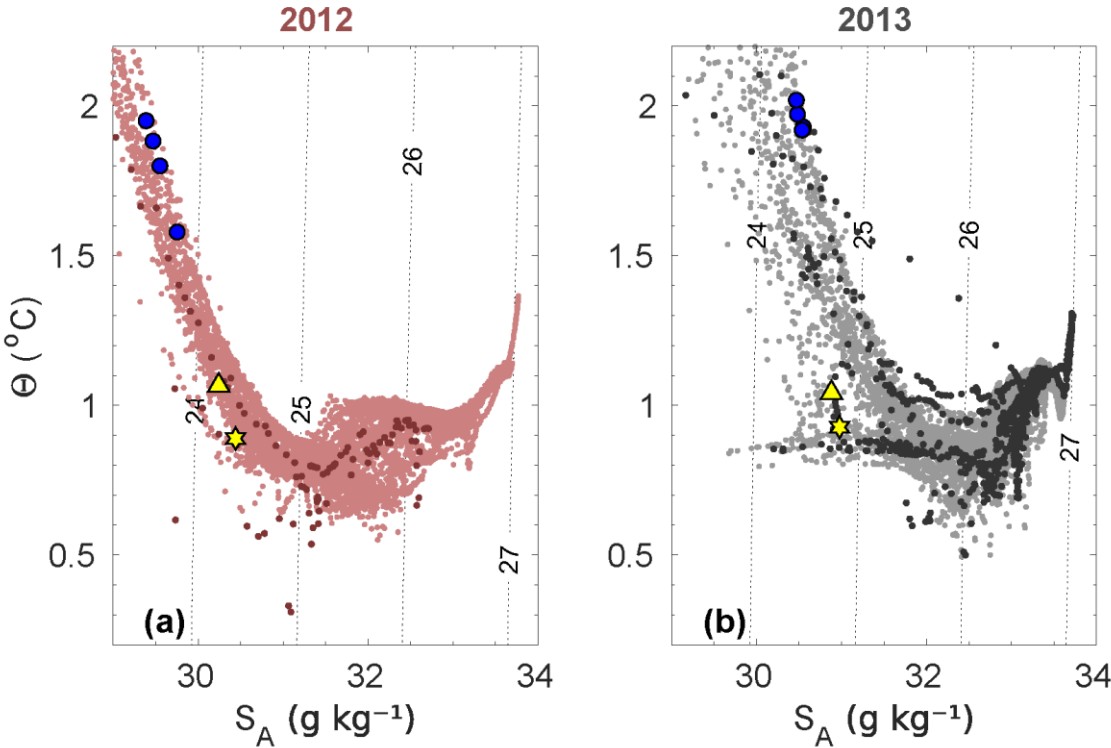

**Figure 10.** Conservative temperature and Absolute salinity of Saqqarleq Fjord waters in a) July 2012 and b) July 2013. Light points show CTD measurements while dark dots are xCTD measurements (closest to the plume). Conservative temperature and absolute salinity at the NBD and MHD as predicted by the plume model are shown as a yellow star and triangle, respectively. The blue solid circles represent the water properties in the core of the observed jets in Fig. 7.

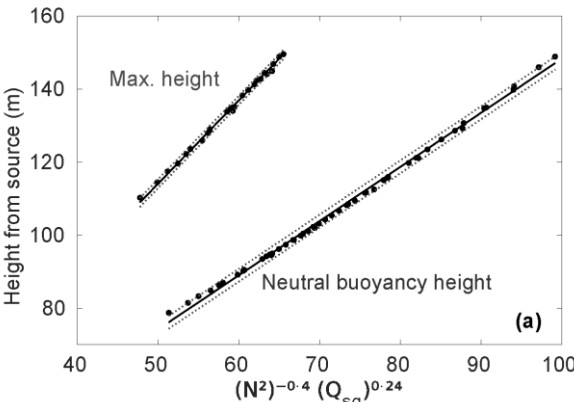
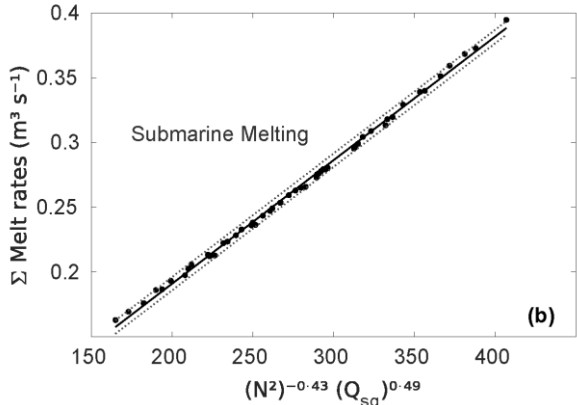

**Figure 11.** Scaling for: a) characteristic plume heights and b) total submarine melt rates from the source to the neutral buoyancy height. Plume model results are plotted by black dots. Straight and dotted black lines represent the fitting curve of Eq. (2) and 95%-confidence bounds, respectively, whose slope and interval bounds values can be found in Table 1.