# Peer review of "Surface emergence of glacial plumes determined by fjord stratification"

_The Cryosphere, 2019_

## Referee Comment (RC1) · Anonymous Referee #1 · 28 Feb 2020

GENERAL COMMENTS:

Using ocean observations from two successive years in combination with the commonly used buoyant plume model, De Andres et al. (2020) show that increased ocean stratification can explain the absence of a surface plume in a Greenland fjord during a high runoff year. They go on to generalize their results using the model, hypothesizing that due to the sensitivity to stratification, increased meltwater discharge may in fact act to suppress the vertical extent of glacial plumes and limit their surface expression.

In general, I found this to be a well-reasoned and well organized paper. The methodology is appropriate, the line of reasoning is clear, and the limitations of the study are acknowledged. I believe the key point made in this paper is of relevance to many aspects related to tidewater glaciers and glacial fjords, and should therefore be of in-

terest to readers of The Cryosphere. I recommend the publication of this manuscript with some modifications, which I have outlined below.

SPECIFIC COMMENTS:

1.

I missed a discussion of the effect of the choice of entrainment coefficient. Quite a large range of values are used in literature, and I suspect that this parameter might have a strong impact on NBD/MHD.

The value used by the authors is not unreasonable, and I do not suggest that an extensive sensitivity analysis is necessary. However, their choice of 0.09 should be justified, and it should at least be discussed how another choice of entrainment parameter might influence the results.

2.

Section 3.1.1. is very short despite the fact that it provides the key observation motivating the study. It should be extended to provide some additional information, directly or through references: Is there more evidence beside these three photographs for the prescence/absence of a plume? Approximately how far did the plume extend along the glacier and into the fjord when it was observed? Was the glacier terminus was located in approximately the same position during 2012 and 2013?

Without a spatial scale in Figure 4 it is also somewhat difficult for a reader to compare the three photographs - please add some sort of reference to the extent of the plume, and ideally to the scale of the images.

3.

Line 275: Please elaborate on how agreement between model and observation is improved in this study compared to these previous studies.

4.

Section 2.3, which describes the methodology wrt. the plume model, should be made clearer. Please provide appropriate references to easily direct the reader to the exact set of equations and parameter values used here, and state explicitly what exactly is meant by "running the plume model" in Section 2.3.2. Are the equations numerically integrated using the observed T/S profiles as boundary conditions? On Lines 99, 107 and 111 the authors refer to Slater (2016) for a description of the model: however, this paper only explicitly contains the plume equations for a half-conical plume, and furthermore discusses both numerical and analytical solutions.

Please also specify the plume water properties with which their plume model was initialized.

5.

It is stated on Lines 123-125 that Qsg and W are both varied at set intervals. Is it not it the combined value Qsg/W that impacts the model, or do these quantities also come into play individually in some other manner? Please clarify.

6.

As far as I understand, it an assumption of the line plume model that the discharge is distributed over a wide enough area that the side interfaces of the plume can be neglected. Using the line plume model with a 10-m width (Line 124) seems likely to violate this assumption. Please justify the use of a line plume model with W as low as 10 m, or acknowledge this as a limitation of the study.

7.

Section 1: This is an excellent introduction section!

8.

The first paragraph of Section 4.2. should be moved to Results, e.g. merged into Section 3.1.2.

9.

Section 2.3.3. should also briefly state how the integrated melt rate is calculated.

10.

Line 284: Please elaborate on, or provide a reference for, why there is a characteristic "plume distance" that might be approximately equal to the grounding line depth.

11.

Line 360-362: This is not new, please remove or modify to reflect that this is in line with previous studies.

12.

Line 371-377: The modelling experiments seem to suggest that the maximum plume height in July 2012 was only a few meters below the surface - I think this should be acknowledged when the reduced nutrient fluxes to the photic zone are discussed.

13.

Line 406: Please explain here or around Line 235 why the jet might be more diffuse in 2012 (reduced stratification?).

14.

The map figure (Figure 1) should be made visually clearer and perhaps used to clarify the description of the study area (see the comment below). It is a little difficult to differentiate between ocean, lakes and land, as well as between sea-ice covered water and glaciers, in the current figure. If possible, I suggest superimposing coastlines in a distinct color.

15.

It was not entirely clear to me from the figure and the text how far the area referred to as SF in fact extends. The text can be read as meaning that SF extends all the way to
the sill near the opening to JI, but the placement of the SF and TF labels in Figure 1 makes it a little unclear e.g. whether the area between the two sills belongs to SF, TF or to the unlabelled fjord to the right. There also seems to be a discrepancy between the length of SF between line 57 (35 km) and line 61 (15 km). Please clarify.

TECHNICAL CORRECTIONS

There are many inconsistencies in the use of past vs present tense throughout the manuscript - I recommend sticking to one or the other through each section.

—-

Abstract: "Ice Sheet" should be "ice sheet" or "Greenland Ice Sheet".

—-

Line 61: Missing space after "(2019)."

—-

Line 106: "tidewater face" should be "glacier face"?

—-

Section 2.3.3: Please specify that the N2 used in the scaling is the mean stratification of the upper layer (not the entire water column) if that is the case.

—-

Line 170: Should 0.11 s-2 be 0.011 s-2? It should also be clarified that this refers to the *maximum* of the mean N2 profile if that is the case.

—-

Line 192: This sentence should be revised for clarity.

—-

Line 204: please specify: "while it did in July 2013" if that is the case.

—-

Line 217-218: Exponentials should be in superscript.

—-

Line 219: Unclear what is meant by this sentence ("Our goal is to identify the model parameters.."). What model parameters are you referring to exactly? Please clarify.

—-

Line 231: "sigma_theta" should be "theta"?

—-

Line 235: "Properties" should be replaced with e.g. "waters".

—-

Line 247: For clarity, please replace "stratification" with "N2" or " B-V frequency squared", etc..

—-

Line 283: Missing "the" before "plume model".

—-

Apparent discrepancy between Line 310 on one hand and Line 248/Figure 11/Table 1 on the other. The latter say the exponent is 0.24, the former says it is 0.26. Please correct or elaborate.

—-

Line 329: There already seems to be strong variability. Do you mean "increased" variability?

—-

Line 368: The meaning of "reaching the fjord surface less" is not clear. Please revise this sentence and clarify.

—

Line 380: Should be plural: "act as atmospheric $CO_2$ sinks".

—

Figure 2: Please revise the colour scheme used in this figure. It is currently difficult to distinguish the black points from the background in the figure on the right. I suspect the figure on the left would be challenging to colorblind readers. I also recommend labelling the subfigures a and b.

A scale bar should also be added to this figure.

—-

Figure 5 caption, first line: should "density" be "potential density"?

—-

Figure 5: I cannot see the horizontal dashed lines referred to in the label.

—-

Figure 7ab: I assume the contours are isolines of sigma theta? They should be explained in the figure caption.

—

Figure 8: I would suggest replacing the x-axis units (DOY) with dates, as it would make for easier comparison with the rest of the manuscript.

$C_{tot}$ and $C_1$ in the figure caption should be formatted with subscripts.
—-

Figure 9: Please specify exactly what the vertical shaded regions represent (one and two standard deviations? standard deviations for the two different years?).

—-

Figure 11a: Please add some information indicating the location of the surface and the top layer here - e.g. as horizontal lines in the plot or as a second y-axis showing "depth below surface".

---

## Referee Comment (RC2) · Anonymous Referee #2 · 10 Mar 2020

Review of Surface emergence of glacial plumes determined by fjord Stratification by Andrés et al.

Andrés et al present oceanographic observations from two 1-2 week campaigns in July in 2012 and 2013 near a tidewater outlet glacier in SF. They include observations of temperature, salinity and ADCP-data near the terminus and apply a simple plume model to analyse the T/S-distributions in the two periods. They speculate that a larger stratification in the upper 10-20 m in 2012 explain why the plume was not observed at the surface in 2012. This is an interesting study and, in particular, the observations in front of the terminus are important for understanding the near-glacial dynamics. I find the application of a simple model justified for the analysis of the role of stratification and their results supports the hypothesis that increased stratification may prevent the

plume from surfacing. However, this mechanism is not a new finding and I find that the detailed discussion of Eq. 2, as it is presented, only explains a feature with this specific model setup and may not represent a general relationship. Model simulations are presented as "evidence", however they only support the hypothesis and the disregard the many assumptions about the real system in the model. More details about the initial conditions for the model simulations are required, and further information about the ADCP-data would be useful. I explain these comments further below. These comments need to be considered before I can recommend publication in Cryosphere.

Comments: Section 2.1 It is not clear whether data was obtained within the plumes from the XCTD's deployed by helicopter. Previous published studies have shown a significant difference between XCTD-profiles deployed in the center of the plumes and the near-by ambient water. Fig. 5 indicate that no profiles were obtained within the plumes. Please clarify whether data was obtained from within the plumes.

The model investigates the role of stratification and the relation between discharge rates and the neutral level. However, it applies the ambient stratification obtained from CTD-profiles. In relation to the comment above, it has been found that the stratification in the plume is significantly different from the ambient conditions. It is not clear how representative the applied stratification in this study is for the near-plume conditions. An analysis of horizontal gradients towards the plumes observed from the CTD-profiles is needed for assessing this important issue.

Eq 2 and Fig. 11: Fig. 2 implies a scaling depending on a and b. However, the found parameters of a and b does not result in a physical dimension of Eq 2 in accordance with the dimension of Z. Thus, the found relation does not represent a scale of the physical system but is related to the model-parameterisation and the applied parameters. It should be clarified to what extent this relation depends on the applied parameters in thid specific model setup.

Figure 7: This is a very interesting figure. However, information about the tides and

winds during the observational periods are missing.

L 360 "We have provided evidence that surface melting of a marine-terminating glacier, and the associated subglacial discharge, together with the fjord's stratification exert a strong control on the dynamics of subglacial discharge plumes with implications for melting of the glacier face and export of meltwater". I do not consider the model simulation as an "evidence". The model results may support the hypothesis, but the applied model has not been validated against observations. The general model formulation is based on plume theory and it has been applied in several studies, but, as the authors point out, there are several assumptions in the choice of model parameters. Please modify the conclusions accordingly.

L 417: It is concluded: "We found that plume vertical extent is proportional to (ðİŚĄ2)−0.4ðİŚĎðİŚăðİŚŤ. While total submarine melting is proportional to (ðİŚĄ2)-0.43ðİŚĎ0.49ðİŚăðİŚŤ. These highlight the important role played by fjord stratification, and the subglacial discharge flux, in the dynamics and impacts of subglacial discharge plumes." These findings are not based on observations, cf. my previous comment. It should be clarified that these relations are not constrained by data but related to the applied model parameters.

Minor comments: L 71: "No statistical differences were found between CTD/xCTD casts taken on different days . . .". Statistical difference (?) has to be clarified.

L 73. "Temperature and conductivity values are converted to conservative temperature (Θ) and absolute salinity (SA) respectively (IOC, SCOR, and IAPSO 2010) using . . .". The references in parenthesis are not included or explained.

L 231: replace sigma-theta with theta

---

## Author Comment (AC1) · 20 Apr 2020

Point-by-point response to editor and reviewer concerns by corresponding author: E. De Andrés

April 10, 2020

**tc-2019-264:**

**Surface emergence of glacial plumes determined by fjord stratification**

Eva De Andrés, Donald A. Slater, Fiamma Straneo, Jaime Otero, Sarah Das, and Francisco Navarro

https://doi.org/10.5194/tc-2019-264

Dear anonymous Reviewer #1,

On behalf of all authors, I would like to thank you for your detailed and constructive comments. In the following you can find a point-by-point response to your comments. We feel that your insights helped improve the manuscript and we hope that all your concerns have been answered to your satisfaction. We would also like to refer you to the responses to the other reviewer for more improvements and changes to the manuscript.

**SPECIFIC COMMENTS / AUTHOR'S ANSWERS**

**1.** I missed a discussion of the effect of the choice of entrainment coefficient. Quite a large range of values are used in literature, and I suspect that this parameter might have a strong impact on NBD/MHD. The value used by the authors is not unreasonable, and I do not suggest that an extensive sensitivity analysis is necessary. However, their choice of 0.09 should be justified, and it should at least be discussed how another choice of entrainment parameter might influence the results.

Thank you for raising this essential point. We in fact find that the NBD and MHD are relatively insensitive to the value of the entrainment coefficient (Fig. R1). Allowing the entrainment coefficient to take values from $\alpha = 0.07$ to $0.12$, modeled NBD ranged from $z = -21$ to $-29$ m in 2012, and from $z = -13$ to $-17$ m in 2013. For any given value of the entrainment coefficient, NBD is deeper in 2012 than in 2013. For the same range of entrainment coefficient values,

modeled MHD ranged from z = -4 to 0 m in 2012, while it remained at z = 0 m in 2013. In general, higher values of the entrainment coefficient leads to a reduced plume vertical extent because the greater entrainment of deep ambient waters makes the plume denser. We chose α = 0.09 for modeling the plume in the two years, because: 1) that value is within the range empirically obtained for geophysical fluid processes (Carazzo et al., 2008), 2) it is the mean of the two values (0.08 and 0.1) used in previous studies of Saqqarleq Fjord (Stevens et al., 2016; Mankoff et al., 2016), and 3) it provided results that fairly accurately predicted the observed jet depths and plume properties. According to your suggestion, we have added these points and this discussion to section 2.3.1 and L329-334.

[Figure]

**Figure R1**. Sensitivity of modeled characteristic plume heights (neutral buoyancy depth, NBD, and maximum height depth, MHD) to the value of the entrainment coefficient (α), in 2012 (left) and 2013 (right). The black continuous line is the observed density profile averaged from all CTD data taken in Saqqarleq Fjord (except those dropped inside the plume). Coloured continuous lines are modeled plume density while coloured dashed and dotted lines represent modeled NBD and MHD, respectively. The subglacial runoff is held constant at the values used in the main paper ($Q_{sg}$ = 101.7 m³/s in 2012 and $Q_{sg}$ = 101.9 m³/s in 2013).

**2.** Section 3.1.1. is very short despite the fact that it provides the key observation motivating the study. It should be extended to provide some additional information, directly or through references: Is there more evidence beside these three photographs for the presence/absence of a plume?

Unfortunately, we cannot further classify the presence or absence of a plume during the field campaigns from satellite imagery. The available images during or immediately around the field campaigns are Landsat 7 or 8 images on July 14 & August 6, 2012 and July 24 & August 2,

2013. None of these images allow us to say anything about the plume, either because of clouds, the small scale of the plume relative to the resolution of the imagery, or stripes in the images (e.g. Fig. R2). We could look at other time periods, but we would rather keep the focus of the paper on the field campaigns when we have concurrent oceanographic observations.

We have other serendipitous photographs of the plume presence/absence taken during fieldwork, but these are no better than those shown in Fig. 4 of the paper. However, we have rearranged and included new photographs in Fig. 4 to facilitate the plume observation. We know from the field surveys that there  was no plume for the duration of the 2012 campaign and that there was a continuous surfacing plume in 2013. We hope the reviewer will find the new existing photos and these statements sufficient evidence for the presence/absence of a plume.

Even though we have not been able to add significant further evidence on the presence/absence of a plume, we have included more explicit and descriptive statements on the presence/absence of the plume, and better described the appearance of the plume in 2013.

[Figure]

**Figure R2**. Satellite imagery of SF/SS on July 14, 2012 (left) and August 6, 2012 (right).

Approximately how far did the plume extend along the glacier and into the fjord when it was observed?

In 2013, the plume extended approximately 200 m parallel to the glacier front and 300 m into the fjord (Mankoff et al, 2016). This has been inserted into the text, L171-172.

Was the glacier terminus located in approximately the same position during 2012 and 2013?

Yes. Multiple terminus position traces for 2012 and 2013 may be found in Fig. 2b of Stevens et al. (2016). This statement has been added to the text, L67-69.

Without a spatial scale in Figure 4 it is also somewhat difficult for a reader to compare the three photographs - please add some sort of reference to the extent of the plume, and ideally to the scale of the images.

We agree that it was difficult to compare the three photographs in Fig.4. Therefore, we have modified Fig. 4 (also included are new photographs) to facilitate the plume comparison between the two years, 2012 and 2013. We have also described the approximate size of the plume surface extent in lines 171-172 and in the caption of the figure.

**3.** Line 275: Please elaborate on how agreement between model and observation is improved in this study compared to these previous studies.

Thank you for this suggestion. In our study, the modeled plume properties at NBD fall within the range of the observed water properties. In both Stevens et al. (2016) and Mankoff et al. (2016), modeled plume properties were consistently too fresh and therefore too light. We attribute our improved model to observation agreement to our use of a line plume model of appropriate width (Jackson et al., 2017), which leads to greater entrainment of denser deep fjord waters than would be achieved with the half-cone plume models used by Stevens et al. (2016) and Mankoff et al. (2016). These points have been added to the manuscript, lines 311-317.

**4.** Section 2.3, which describes the methodology wrt. the plume model, should be made clearer. Please provide appropriate references to easily direct the reader to the exact set of equations and parameter values used here, and state explicitly what exactly is meant by "running the plume model" in Section 2.3.2. Are the equations numerically integrated using the observed T/S profiles as boundary conditions? On Lines 99, 107 and 111 the authors refer to Slater (2016) for a description of the model: however, this paper only explicitly contains the plume equations for a half-conical plume, and furthermore discusses both numerical and analytical solutions. Please also specify the plume water properties with which their plume model was initialized.

We have not described the plume model in great detail in our paper because it has become a very standard tool in the related literature (e.g. Slater et al., 2015, 2016, 2017; Jenkins, 2011; Stevens et al., 2016; Mankoff et al., 2016; Jackson et al., 2017; Carroll et al., 2016), but we have now improved the description by following all of the reviewer's suggestions. The equations are indeed numerically integrated from the grounding line to the MHD or fjord surface, whichever comes first, using the observed T/S profiles as boundary conditions (now stated on lines 118-120). Slater et al. (2016) do use the numerical line plume model also used in our paper, but you are right that the equations are not explicitly stated; these can instead be found in Jenkins

(2011) as now stated on lines 103 and 110. The plume model is initialised with the flux of subglacial discharge (the magnitude of which is described later in the paper), with this discharge assumed to have zero salinity and to be at the pressure melting point. These details have been added on line 119-120.

**5.** It is stated on Lines 123-125 that Qsg and W are both varied at set intervals. Is it not it the combined value Qsg/W that impacts the model, or do these quantities also come into play individually in some other manner? Please clarify.

Yes, the reviewer is correct that the only dynamically-relevant quantity is Qsg/W - these quantities do not appear individually anywhere in the line plume model. This is why we are able to plot characteristic heights versus Qsg/W in Fig. 9 of the article. Our reason for describing Qsg and W separately in some places in the paper is to make contact with the real system (in which clearly Qsg has a value in $m^3$/s defined by surface melting of the glacier and W is set by the dynamics of subglacial channels). In doing so we hope to make the paper more accessible. Therefore we would rather leave these lines as they are, but we have added a clarification that the only quantity that enters the model is Qsg/W (lines 135-136).

**6.** As far as I understand, it is an assumption of the line plume model that the discharge is distributed over a wide enough area that the side interfaces of the plume can be neglected. Using the line plume model with a 10-m width (Line 124) seems likely to violate this assumption. Please justify the use of a line plume model with W as low as10 m, or acknowledge this as a limitation of the study.

We thank the reviewer for raising this important point. We agree and have increased the minimum channel width we consider from 10 to 50 m (line 135). In fact this doesn't change any of our plots because the new range of Qsg/W ratios still covers the ranges that were previously plotted (e.g. on Fig. 9).

**7.** Section 1: This is an excellent introduction section!

Thank you.

**8.** The first paragraph of Section 4.2. should be moved to Results, e.g. merged into Section 3.1.2.

Following your recommendation, the first paragraph of section 4.2 has been merged into section 3.1.2 (L178-186).

**9.** Section 2.3.3. should also briefly state how the integrated melt rate is calculated.

The submarine melt rate, *m*, at a point on the calving front within the plume is calculated using the plume model. This does not vary within the width (*W*) of the plume, and therefore the integrated melt rate is defined as

$$M = W \int_{z=-150}^{z=-NBD} m(z)dz$$

We have added this information on L159-161 and in the new Eq. (3).

**10.** Line 284: Please elaborate on, or provide a reference for, why there is a characteristic "plume distance" that might be approximately equal to the grounding line depth.

This section describes how significant mixing occurs as waters from the plume flow horizontally away from the glacier close to the fjord surface, but including reference to a specific 'plume distance' here is unnecessary and so we have changed this to 'a few hundred metres' (line 324).

**11.** Line 360-362: This is not new, please remove or modify to reflect that this is in line with previous studies.

Modified as suggested (lines 405-407).

**12.** Line 371-377: The modelling experiments seem to suggest that the maximum plume height in July 2012 was only a few meters below the surface - I think this should be acknowledged when the reduced nutrient fluxes to the photic zone are discussed.

Agreed - the model is suggesting that while the plume was not observed to surface in July 2012, it must have been very close to surfacing. We have now acknowledged in lines 417-419 and 424-425 that the impact on vertical nutrient fluxes in SF in July 2012 may have been small.

**13.** Line 406: Please explain here or around Line 235 why the jet might be more diffuse in 2012 (reduced stratification?).

On L211-214, we have included the potential explanation and the observed values of $N^2$ at the jet depth in each year as support for the hypothesis.

**14.** The map figure (Figure 1) should be made visually clearer and perhaps used to clarify the description of the study area (see the comment below). It is a little difficult to differentiate between ocean, lakes and land, as well as between sea-ice covered water and glaciers, in the current figure. If possible, I suggest superimposing coastlines in a distinct color.

Great advice. We have changed Figure 1 according to the reviewer's suggestions.

**15.** It was not entirely clear to me from the figure and the text how far the area referred to as SF in fact extends. The text can be read as meaning that SF extends all the way to the sill near the opening to JI, but the placement of the SF and TF labels in Figure 1 makes it a little unclear e.g. whether the area between the two sills belongs to SF, TF or to the unlabelled fjord to the right. There also seems to be a discrepancy between the length of SF between line 57 (35 km) and line 61 (15 km). Please clarify.

Apologies that the text was rather unclear here. We consider SF to extend up to the 70 m deep sill ~15 km from SS; beyond this is TF extending up to the junction with JI. We have reworded the first paragraph of section 2 and revised Fig. 1 in line with the reviewer's comment above to clarify these points.

**TECHNICAL CORRECTIONS**

There are many inconsistencies in the use of past vs present tense throughout the manuscript - I recommend sticking to one or the other through each section.

Thank you for this suggestion - we have reviewed the tenses and now consistently use the past tense in sections describing the observations and the present tense when describing the plume model and model results (e.g. see changes on line 77, 96, 187-192, 271-272, 301, 304, 321, etc).

Abstract: "Ice Sheet" should be "ice sheet" or "Greenland Ice Sheet".

Changed as suggested.

Line 61: Missing space after "(2019)."

Added.

Line 106: "tidewater face" should be "glacier face"?

Yes, changed as suggested.

Section 2.3.3: Please specify that the N2 used in the scaling is the mean stratification of the upper layer (not the entire water column) if that is the case.

Yes, $N^2$ in the scalings is the stratification of the top layer. This has been clarified in the text (lines 153-154).

Line 170: Should 0.11 s-2 be 0.011 s-2? It should also be clarified that this refers to the *maximum* of the mean N2 profile if that is the case.

Changed and clarified as suggested.

Line 192: This sentence should be revised for clarity.

Revised.

Line 204: please specify: "while it did in July 2013" if that is the case.

Done.

Line 217-218: Exponentials should be in superscript.

Now corrected.

Line 219: Unclear what is meant by this sentence ("Our goal is to identify the model parameters.."). What model parameters are you referring to exactly? Please clarify.

Essentially we are considering whether the plume model can reproduce the observations, and have changed the sentence accordingly.

Line 231: "sigma_theta" should be "theta"?

Corrected.

Line 235: "Properties" should be replaced with e.g. "waters".

Replaced.

Line 247: For clarity, please replace "stratification" with "N2" or " B-V frequency squared", etc..

Replaced with $N^2$.

Line 283: Missing "the" before "plume model".

Added.

Apparent discrepancy between Line 310 on one hand and Line 248/Figure 11/Table 1 on the other. The latter say the exponent is 0.24, the former says it is 0.26. Please correct or elaborate.

The correct value is 0.24. We have corrected the mistake on line 365 (old L310) - thank you for spotting this.

Line 329: There already seems to be strong variability. Do you mean "increased" variability?

In this paragraph we mean to highlight variability within a year (i.e. seasonal variability), whereas previously in the paper we have mainly contrasted 2012 and 2013 (i.e. interannual variability, now explicitly included in L376 )

Line 368: The meaning of "reaching the fjord surface less" is not clear. Please revise this sentence and clarify.

Revised: "plumes may reach the fjord surface less **often** over the coming century"

Line 380: Should be plural: "act as atmospheric CO2 sinks".

Corrected.

Figure 2: Please revise the colour scheme used in this figure. It is currently difficult to distinguish the black points from the background in the figure on the right. I suspect the figure on the left would be challenging to colorblind readers. I also recommend labelling the subfigures a and b. A scale bar should also be added to this figure.

Figure 2 has been changed following all of your suggestions.

Figure 5 caption, first line: should "density" be "potential density"?

Yes. Corrected.

Figure 5: I cannot see the horizontal dashed lines referred to in the label.

We have now added the horizontal dashed lines.

Figure 7ab: I assume the contours are isolines of sigma theta? They should be explained in the figure caption.

Explanation added to the caption.

Figure 8: I would suggest replacing the x-axis units (DOY) with dates, as it would make for easier comparison with the rest of the manuscript. Ctot and C1 in the figure caption should be formatted with subscripts.

We would prefer to keep the DOY, as it makes our study easily comparable to the two previous studies in SF (Mankoff et al., 2016; Stevens et al., 2016), but we have added a conversion of DOY to the field survey dates in the figure caption. We have also formatted the subscripts.

Figure 9: Please specify exactly what the vertical shaded regions represent (one and two standard deviations? standard deviations for the two different years?).

The regions represent one standard deviation of the subglacial runoff during the 5 day period preceding the velocity measurements in the fjord. This has been clarified in the figure caption.

Figure 11a: Please add some information indicating the location of the surface and the top layer here - e.g. as horizontal lines in the plot or as a second y-axis showing "depth below surface".

The interface between the top and bottom layer is 100 m below the surface and is therefore outside of the y-axis limits shown in Fig. 11a (all tested plumes reach significantly higher than this interface). We have added the fjord surface level to Fig. 11a.

---

## Author Comment (AC2) · 20 Apr 2020

Point-by-point response to editor and reviewer concerns by corresponding author: E. De Andrés

April 10, 2020

**tc-2019-264:**

**Surface emergence of glacial plumes determined by fjord stratification**

Eva De Andrés, Donald A. Slater, Fiamma Straneo, Jaime Otero, Sarah Das, and Francisco Navarro

https://doi.org/10.5194/tc-2019-264

Dear anonymous Reviewer #2,

We would like to thank the reviewer for their helpful and constructive comments, and for taking the time to review our manuscript. In the following we give a point-by-point response to each comment and hope that the reviewer finds the manuscript to be improved. We would also like to refer you to the responses to the other reviewer for more improvements and changes to the manuscript.

**COMMENTS / AUTHOR'S ANSWERS**

**1.** Section 2.1 It is not clear whether data was obtained within the plumes from the XCTD's deployed by helicopter. Previous published studies have shown a significant difference between XCTD-profiles deployed in the center of the plumes and the near-by ambient water. Fig. 5 indicates that no profiles were obtained within the plumes. Please clarify whether data was obtained from within the plumes.

Thanks for pointing out this important issue. In 2013, the 12 xCTDs deployed by the helicopter entered the water within the surface expression of the plume (Fig. 4), as did 8 of the CTD casts from the boat. Because the rising core of the plume is likely narrow and confined against the ice, these casts may not have stayed inside the plume all the way to the sea floor. These 'in-plume' casts have been extensively described and analysed by Mankoff et al. (2016); see e.g. their Figure 5, which indeed shows a significant difference in properties inside and outside of the

plume. During the 2012 campaign, just 2 xCTDs were deployed by the helicopter and, although not certain (due to the lack of a plume surface expression), they seemed to fall within or near the plume pool.

In Fig. 5, we don't include any casts from within the plume because we wish to highlight the ambient waters that are providing the boundary conditions for the rise of the plume. In Fig. 10, we do include casts from inside the plume because we are comparing these observations to the output of the plume model.

We have clarified these points on lines 73, 132-134, and in the caption of Fig. 5.

**2.** The model investigates the role of stratification and the relation between discharge rates and the neutral level. However, it applies the ambient stratification obtained from CTD-profiles. In relation to the comment above, it has been found that the stratification in the plume is significantly different from the ambient conditions. It is not clear how representative the applied stratification in this study is for the near-plume conditions. An analysis of horizontal gradients towards the plumes observed from the CTD-profiles is needed for assessing this important issue.

First, it is important to point out that the boundary conditions for the plume model should be the ambient waters through which the plume is rising. The stratification within the plume itself is what the plume model is trying to simulate and therefore should not be used to set the ambient/boundary conditions for the plume model.

The reviewer is correct however that the ambient conditions felt by the plume, presumably the fjord waters very close to the plume but not inside it, might differ from those further away (say a few km from the front). Spatial variability in SF water properties has been analysed for 2012 and 2013 by Stevens et al. (2016) and Mankoff et al. (2016), respectively, and so we do not think it would be appropriate to include a similar analysis in our manuscript. We have, however, conducted an analysis of how sensitive our plume model results are to how we set our ambient stratification in each year.

For this analysis, following Stevens et al. (2016), we have grouped CTD casts within 150 m of the calving front and close to the plume (150 m - D1), casts within 150 m of the calving front but at the other side of the calving front (150 m - D2), and casts along the velocity transect ~1.5 km from the calving front (1500 m - R1). We then ran the plume model in each year with ambient conditions defined using these 3 groups of CTD casts and realistic subglacial runoff (Fig. R3). In 2012, NBD varied between 20 and 26 m, and in 2013 between 13 and 15 m depending on the ambient conditions used. MHD ranges from 0 to 3 m in 2012 and is always 0 m in 2013.

These tests show NBD/MHD are quite insensitive to how we define our ambient conditions. This may already have been guessed based on the small differences between ambient CTD profiles taken at different points in the fjord (Fig. R3). Most importantly, the characteristic plume heights

are deeper in 2012 than in 2013 for any definition of the ambient conditions (Fig. R3). For the results in our manuscript and for simplicity, we decided to prescribe ambient conditions for the plume model as the average over all CTD casts in each year, excluding those from within the plume. Fig. R3 shows that this definition is sufficient.

We have now clarified how we define the ambient conditions for the plume model (L118-119, 242-243 and caption of Fig. 5) and included a discussion of these sensitivity tests in L334-342.

[Figure]

**Figure R3**. Sensitivity of modeled characteristic plume heights to the ambient water properties observed at different distances from the front, in 2012 (left) and 2013 (right). Coloured continuous lines are modeled plume and ambient density (as indicated) while coloured dashed and dotted lines represent modeled NBD and MHD, respectively. D1 and D2 are main and secondary plume locations, respectively (defined in Stevens et al., 2016). Subglacial runoff is held constant at the values used in the main paper ($Q_{sg}$ = 101.7 m³/s in 2012 and $Q_{sg}$ = 101.9 m³/s in 2013).

**3.** Eq 2 and Fig. 11: Fig. 2 implies a scaling depending on a and b. However, the found parameters of a and b does not result in a physical dimension of Eq 2 in accordance with the dimension of Z. Thus, the found relation does not represent a scale of the physical system but is related to the model-parameterisation and the applied parameters. It should be clarified to what extent this relation depends on the applied parameters in this specific model setup.

We have now revised both the height and the melt rate scalings so that the constant A and the quantities raised to the powers a and b are all dimensionless (section 2.3.3). With this recasting, the relation does contain fundamental scales of the physical system and it is clear how model parameters affect the scalings, but we have not had to change any of our analysis or results (e.g.

Fig. 11). To avoid overcomplicating the manuscript at this point, we have placed details of the scalings in a new appendix (Appendix A).

**4.** Figure 7: This is a very interesting figure. However, information about the tides and winds during the observational periods are missing.

The amplitude of the barotropic and baroclinic tidal currents, derived from an ADCP deployed in the middle of the fjord in summer of 2012, are approximately 0.01 m/s and 0.06 m/s respectively (R.M. Sanchez, personal communication). These currents are much smaller than those observed in the jet, ~ 0.3 m/s and shown in Fig. 7 , thus we do not expect that removal of the tidal velocities would significantly change the structure of the jet. The jet structure, in turn, is used mostly to identify the water masses that are carried away from the glacier in the jet.

Unfortunately, no local wind observations are available for the duration of the 2012 and 2013 surveys. During both surveys, however, wind conditions and sea-state were largely calm and permitted surveys to be conducted from small boats and autonomous vehicles. This observation, together with the highly localized nature of the jet, support the conclusion that the jet is associated with subglacial discharge plume, and is not a wind-driven feature. The numerical simulations of Slater et al. 2018, who are able to reproduce the jet with no wind forcing, support this conclusion.

Following your comments, we have included this information in section 3.1.3 (L217-226).

**5.** L360 "We have provided evidence that surface melting of a marine-terminating glacier, and the associated subglacial discharge, together with the fjord's stratification exert a strong control on the dynamics of subglacial discharge plumes with implications for melting of the glacier face and export of meltwater". I do not consider the model simulation as an "evidence". The model results may support the hypothesis, but the applied model has not been validated against observations. The general model formulation is based on plume theory and it has been applied in several studies, but, as the authors point out, there are several assumptions in the choice of model parameters. Please modify the conclusions accordingly.

We respectfully disagree with the reviewer on this particular point. First, this statement is not based solely on the model simulation. Figs. 4, 7 and 10 provide observations of differing plume and jet dynamics in 2012 and 2013, and we have attributed this to differing fjord stratification, which is also observed. The plume model is used to provide a dynamical understanding of our observations. Second, we do think that the model has been (at the very least partly) validated against observations; as described in section 3.3 (i) modeled NBD is close to the observed jet depth, (ii) modeled MHD matches the photographs of plume-patch presence/absence, and (iii) modeled plume T-S properties at NBD and MHD are close to xCTD observed T-S properties. Third, although there are several assumptions in the choice of model parameters and boundary

conditions, our results are largely insensitive to these choices (e.g. the entrainment coefficient - see new lines 328-333, and the ambient conditions - L334-342, see response to comment above).

We do think, therefore, that the highlighted statement and our conclusions are warranted, but we have reworded this statement (line 405) to emphasize that we are basing this statement on both observations and the plume model, with the plume model used to provide a dynamical understanding of why the plume differed in the two years.

The reviewer is of course correct that the scalings we derive are based purely on the plume model (although our observations and numerous other applications in the literature provide support that the plume model is sensible). We have now made it clear in the conclusions that the scalings are based on the plume model and not the observations (see response to next comment).

**6.** L417: It is concluded: "We found that plume vertical extent is proportional to $(N^2)^{-0.4}Q_{sg}^{0.24}$, while total submarine melting is proportional to $(N^2)^{-0.43}Q_{sg}^{0.49}$. These highlight the important role played by fjord stratification, and the subglacial discharge flux, in the dynamics and impacts of subglacial discharge plumes." These findings are not based on observations, cf. my previous comment. It should be clarified that these relations are not constrained by data but related to the applied model parameters.

We agree - it is important to make clear the distinction between the general point of plume dynamics being affected by fjord stratification (which is evidenced by our observations and supported by the plume model), and the quantitative scalings (which are based only on the plume model). And since they are based on the plume model, the quantitative scalings are indeed related to the applied model parameters. These points have now been made clear in the conclusions (section 5).

**MINOR COMMENTS / AUTHOR'S ANSWERS**

L71: "No statistical differences were found between CTD/xCTD casts taken on different days...". Statistical difference (?) has to be clarified.

The mean temperature and salinity among CTD casts taken on different days are not statistically different since a one-way ANOVA (ANalysis Of VAriance) indicates p > 0.05. This has been clarified on L75.

L73. "Temperature and conductivity values are converted to conservative temperature ($\Theta$) and absolute salinity (SA) respectively (IOC, SCOR, and IAPSO 2010) using...". The references in parenthesis are not included or explained.

Reference included in L577 (Reference list).

L231: replace sigma-theta with theta.

Replaced - thank you for spotting this mistake.